# A Semi-Physical Approach for Downscaling Satellite Soil Moisture Data in a Typical Cold Alpine Area, Northwest China

**Zetao Cao [1], Hongxia Gao [1], Zhuotong Nan [1,2,*], Yi Zhao [1] and Ziyun Yin [1]**

[1] Key Laboratory of Ministry of Education on Virtual Geographic Environment, Nanjing Normal University, Nanjing 210023, China; caozt@nnu.edu.cn (Z.C.); 161302060@njnu.edu.cn (H.G.); zhaoyi030@njnu.edu.cn (Y.Z.); yinziyun0208@nnu.edu.cn (Z.Y.)

[2] Jiangsu Center for Collaborative Innovation in Geographical Information Resource Development and Application, Nanjing 210023, China;

\* Correspondence: nanzt@njnu.edu.cn

**Abstract:** Microwave remote sensing techniques provide a direct measurement of surface soil moisture (SM), with advantages for all-weather observations and solid physics. However, most satellite microwave soil moisture products fail to meet the requirements of land surface studies for high-resolution surface soil moisture data due to their coarse spatial resolutions. Although many approaches have been proposed to downscale the spatial resolution of satellite soil moisture products, most of them have been tested in flat areas where the surface is relatively homogeneous. Thus, those established approaches are often inapplicable for downscaling in cold alpine areas with complex terrain where multiple factors control the variations in surface soil moisture. In this work, we re-inferred and verified the mathematical assumption behind a semi-physical approach for downscaling satellite soil moisture data and extended this approach for cold alpine areas. Instead of directly deriving SM from proxy variables, this approach relies on a relationship between two standardized variables of SM and apparent thermal inertia (ATI), in which the sub grid standard deviation for SM is estimated by a physical hydraulic model taking soil texture data as input. The approach was applied to downscale the soil moisture active passive (SMAP) daily data in a typical cold alpine basin, i.e., the Babao River basin located in the Qilian Mountains of Northwest China. We observed good linearity between the computed ATI and SM observations on most wireless sensor network sites installed in the study basin, which justifies the underlying assumption. The sub grid standard deviations for the SMAP grid estimated through the Mualem-van Genuchten model can broadly represent the real characteristics. The downscaled 1-km resolution results correlated well with the in-situ SM observations, with an average correlation coefficient of 0.74 and a small root mean square error (0.096 cm³/cm³). The downscaled results show more and consistent textural details than the original SMAP data. After removal of biases in the original SMAP data even higher agreements with the observations can be achieved. These results demonstrate the adequacy of the proposed semi-physical approach for downscaling satellite soil moisture data in cold alpine areas, and the resultant fine-resolution data can serve as useful databases for land surface and hydrological studies in those areas.

**Keywords:** soil moisture; statistical downscaling; semi-physical approach; cold alpine area; sub grid variability; apparent thermal inertia

## 1. Introduction

Surface soil moisture (SM), defined as the relative water content of the top few centimeters soil (up to 5 cm depth), is a crucial variable in terrestrial and atmospheric water cycles [1,2]. It is a storage component for precipitation and radiation, controlling the partitioning of precipitation into surface runoff and infiltration and the partitioning of avail-

able energy into sensible and latent heat fluxes. SM can affect the water, energy and bio-geochemical cycles by influencing plant transpiration and photosynthesis. It also participates in the feedbacks of the underlying surface to atmosphere at local, regional and global scales and plays an important role in climate-change projections [3,4]. SM has strong spatial heterogeneity, which makes typical field surveys incapable of capturing the SM dynamics in large areas [5,6]. In recent years, wireless sensor networks have become an important technique to measure real-time SM data in the field, with the advantages of high accuracy in capturing spatial and temporal dynamics of SM at small regions [7,8]; however, these approaches are not applicable for SM observations over wide areas. Compared with ground observations, remote sensing has the strength of providing SM observations from regional to global scales. The synthetic aperture radars are characterized by a high spatial resolution through the emission and reception of electromagnetic signals, but their temporal resolutions are often an issue for regional studies [9–11]. Their Coarse-resolution microwave radiometers and scatterometers operating at L-band has become a major approach to monitor SM [3,12,13]. They provide frequent revisit times at a grid resolution of tens of kilometers (AMSR-E 25 km, SMOS 40 km, SMAP 36 km/9 km), too coarse for land surface and hydrological applications in meso- and small-scale studies [1,2].

Many downscaling approaches have been proposed to downscale the coarse spatial resolution of satellite SM products [1,2]. These approaches can be classified as statistical and dynamical approaches. Statistical downscaling approaches generally downscale coarse-scale SM products using modulating variables at finer spatial resolution by establishing empirical relations between them. Since SM impacts surface turbulent energy fluxes, the surface radiant temperature is sensitive to SM with low vegetation cover. Based on this, Carlson et al. [14,15] quantified the relationships among SM, surface radiant temperature and vegetation and then proposed a triangle method. An empirical polynomial fitting method based on this relationship was then developed and validated in the Murrumbidgee catchment in south-eastern Australia and a region of the Great Plain in the USA [16,17]. Based on the "universal triangle" method, Xu et al. [18] proposed a new approach that fuses SM data with Landsat 8 and Moderate Resolution Imaging Spectroradiometer (MODIS) datasets. Remote sensing indices related to vegetation, humidity and temperature have also been employed in statistical downscaling approaches [19–22] to improve the performance. Due to the large heat capacity of water, SM greatly affects the ability of soil to resist temperature changes, and this property is usually expressed as thermal inertia. Soil thermal inertia, which is modulated by various factors (SM, topography, vegetation, soil texture, etc.), can reflect the synthetic properties of the soil and has a high potential in statistically downscaling SM data [23,24]. However, estimating real thermal inertia requires accurate soil parameters such as thermal conductivity, bulk density, and specific heat capacity, which are hard to obtain for a large area. In practical studies, apparent thermal inertia (ATI), which represents the relative value of thermal inertia and can be estimated through remote sensing techniques, is often regarded as an approximation of thermal inertia [25,26]. As ATI shows good consistency with SM at site scale, especially in poorly vegetated areas [27], it has been widely used in SM retrieval and downscaling studies [25–30]. However, in large areas with complex underlying surfaces where SM is jointly controlled by many factors, the statistical relationship between SM and any single environmental variable is subject to great spatiotemporal heterogeneity, limiting their applicability in those areas. In recent years, with advances in computer techniques, machine learning has been more often employed in SM downscaling studies [31–33]. The performance of using machine learning appears to be satisfactory when ample training samples are available.

Statistical downscaling approaches are advantageous of simplicity, but it depends on prior knowledge; hence, it is more applicable to the areas that are relatively homogeneous and with abundant data. Dynamical downscaling approaches usually are based on physical models. In spite of more complexity, they do not require much in situ data and can be applied to ungauged areas where statistical relationships are hard to establish. Merlin et

al. [34] built a dynamical downscaling approach (the DISPATCH method) that models soil evaporation processes. Later, vegetation cover, soil type, and atmospheric conditions were added to improve the accuracy in calculating soil evaporation efficiency [35,36]. The method has been tested with different SM products and successfully applied in several regions [37,38]. Some hydrological or land surface models (VIC, Noah, etc.) have also been employed to assimilate and downscale the SM [39,40]. However, although the dynamical approach differs from the statistical one in the nature of methodology, no obvious superiority has been observed in terms of downscaling accuracy. Therefore, in practice, a most appropriate approach needs to be determined according to natural characteristics of the study area and the situation of data availability [41].

Most study areas in SM downscaling research have low altitudes and are relatively flat [16–18,24,38–40]. However, cold alpine areas are characterized by strong land-surface heterogeneity and high nonlinearity between SM and environmental variables. The statistical relationships established for low-altitude and flat areas are usually not applicable to the cold alpine areas. In such areas, SM is susceptible to multiple factors, including precipitation [42], topography [43], and vegetation [44], making it impossible to single out a dominant factor from those variables. The interactions between SM and environmental factors become even more complicated due to the presence of glaciers and permafrost that are unique in cold alpine areas [45]. In this sense, physics-based downscaling approaches following physical laws may have advantages over pure statistical approaches in dealing with strong heterogeneity in cold alpine areas. Qu et al. [46] derived a closed-form expression to describe SM variability using stochastic analysis of one-dimension unsaturated gravitational flow based on the Mualem-van Genuchten (MvG) model [47] fed with soil texture data. On this basis, Montzka et al. [48] used Qu's physical method [46] to predict sub grid SM standard deviations and established a semi-physical/semi-empirical downscaling approach for SM, with a case study in the Upper Rhine Valley, Germany. In theory, such semi-physical approach combining physical and empirical methods might be promising in downscaling SM in cold alpine areas since physical methods are good at dealing with strong heterogeneity while the simplicity of empirical methods is still retained.

However, there is a flaw in Montzka's study [48], which assumes a statistical relationship between SM and field capacity maintaining at both fine and coarse resolutions. Generally, field capacity is a constant soil property and will not change in a short period. While SM shows considerable variabilities in time, the constant field capacity fails to reflect these variabilities. In fact, in cold alpine areas, vegetation, topography, or orographic precipitation alone could not be dominant factors of SM [42–44], nor could they be used as an effective proxy variable for downscaling. This study extends and modifies Montzka's approach in attempt to establish an applicable semi-physical SM downscaling approach for cold alpine areas. In this hybrid approach, a physical model is employed to predict sub grid standard deviations for Soil Moisture Active Passive (SMAP) SM grid cells, and a statistical relation built upon sub grid standard deviations of SM and ATI is used to downscale the SMAP SM data into 1-km resolution. The approach was evaluated with the in-situ SM observations obtained from a wireless sensor network in the Qilian Mountains [49], a typical cold alpine area in Northwest China.

## 2. Study Area and Data

### 2.1. Study Area

The Babao River basin (BRB) is located upstream of the Heihe River, the second-largest inland river in China (Figure 1a), with latitudes of 37°43′ N~38°20′ N and longitudes of 100°05′ E~101°09′ E. The length of the mainstream is 101 km, with a drainage area of approximately 2452 km². The altitudes range from 2687 to 4960 m with a mean of 3604 m. The BRB is a typical cold alpine area with a mean annual temperature of 0.7 °C, where frozen ground is widely distributed. The lower limit of permafrost is approximately

3650~3700 m. The rainy season is from May to September, with most rainfall occurring in summer (from June to August), and there are a few snowfall events from October to April. The annual precipitation is approximately 300~500 mm. In addition, precipitation increases at elevations below 3650 m and decreases at higher elevations [50]. In the rainy season, precipitation is unevenly distributed due to the impacts of topography and mountain microclimates, which leads to rapid changes and strong spatiotemporal heterogeneity in SM [51]. The land cover exhibits vertical zonality, with alpine meadows and subalpine shrubs being the main vegetation types, and there is a small area of forest at approximately 3100 m. Vegetation has a great impact on the streamflow as well as the regulation and storage of water in the basin [52]. Glaciers and snowpack exist above 4500 m.

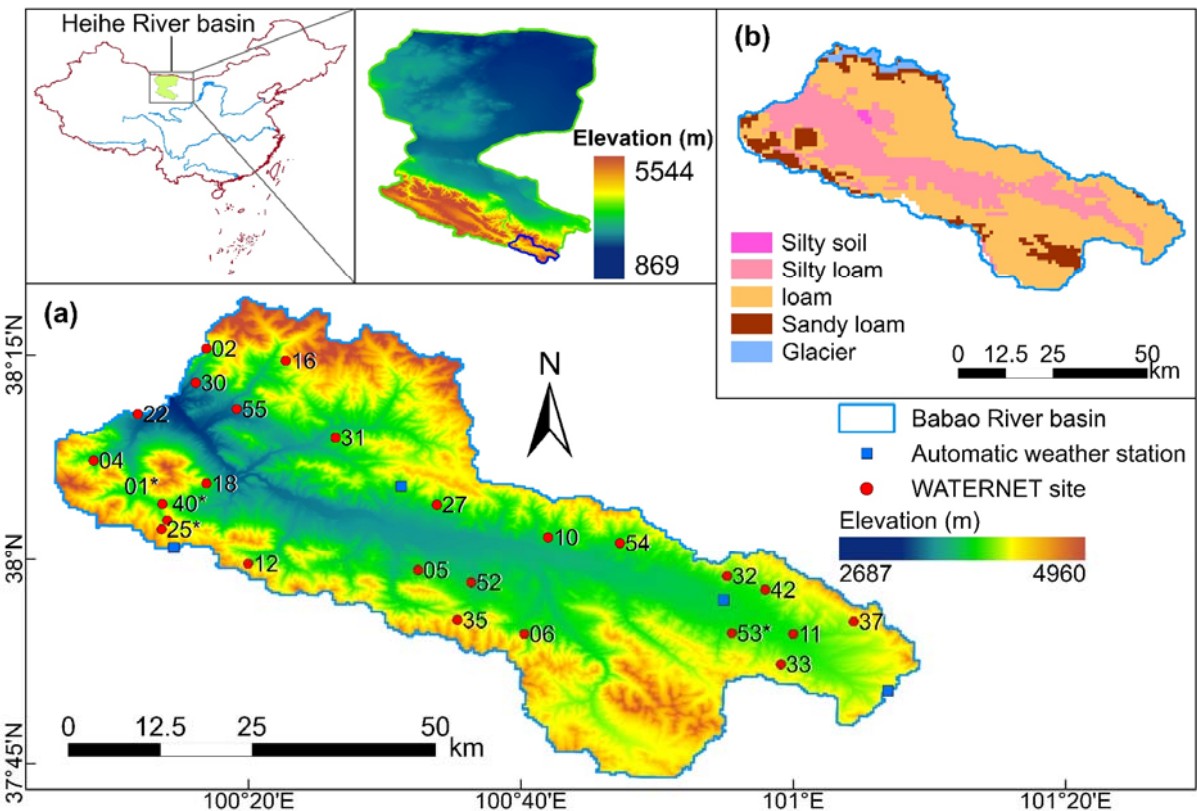

**Figure 1.** Map showing (**a**) the location and topography of the Babao River basin (BRB) and the Watershed Allied Telemetry Experimental Research Net (WATERNET) sites, and (**b**) the soil texture. The numbers marked are the WATERNET site IDs. The asterisk (*) following a site ID indicates the site with compromised downscaling quality using the proposed approach.

### 2.2. SMAP Data

The National Aeronautics and Space Administration (NASA) Soil Moisture Active Passive mission satellite was launched in 2015 with L-band (active) radar and L-band (passive) radiometer equipped. The radar stopped functioning on 7 July 2015 [53]. The basic SMAP mission is to provide high accuracy, high temporal-spatial resolution, and global cover SM data through combined radar and radiometer measurements. The radiometer can measure soil moisture up to 5 cm depth. In spite of its coarse resolution, the SMAP SM data have been used in many studies (e.g., crop yield and irrigation [54], hydrological simulation [55], and weather prediction [56]). The satellite SM data used in this study are SMAP Enhanced L3 Daily Global Composite Radiometer Soil Moisture (SMAP_L3_SM_P_E product), of which the spatial resolution has been downscaled from an original resolution of approximately 33 km to 9 km by the Backus Gilbert optimum interpolation technique [53]. The product includes twice SM measurements at 6 am and 6 pm. Several studies have indicated that SMAP Enhanced L3 data have comparatively high

accuracy and can well represent spatiotemporal variabilities in SM [57–59]. Due to the high number of missing observations in the morning, only the SM measurements in the afternoon were used as the downscaling target in this study. The SMAP products are available at the NASA National Snow & Ice Data Center (https://nsidc.org).

The baseline retrieval algorithm for SMAP L2/L3 passive soil moisture products determines the surface frozen/non-frozen state by the effective soil temperature modelled by the Goddard Earth Observing System Model forward processing of Global Modeling and Assimilation Office (GMAO GEOS-FP), NASA. A frozen soil fraction, defined as the proportionality of frozen cells, is generated to guide the SMAP operational processor. If frozen soil fraction in a cell is 0.50–1.00, the cell is flagged as frozen ground and no SM is retrieved [60]. Otherwise, if a cell has a small portion of frozen ground, i.e., a fraction of 0.05–0.50, the cell is flagged as non-frozen ground and the SM is attempted to retrieve [60]. It means the SMAP still has capability in detecting the SM in the transition periods between cold and warm seasons when partial areas in the cell freeze. As frozen soils have similar dielectric constants to dry soils [61], the SMAP SM excludes the ice content that should be a part of total soil moisture and includes only liquid water. Meanwhile, the in-situ observations are measured by the time-domain reflectometry [49], including only free water. According to the experiment of Zhang et al. [62], the SMAP baseline algorithm can produce comparable results to the in-situ observations when the surface is frozen. Given the importance of frozen ground to cold alpine areas, we do not discard this part of SMAP SM data that may be influenced by frozen ground and will specially evaluate them.

### 2.3. MODIS Land Surface Temperature and Reflectance

In this study, the land surface temperature (LST) product (MOD11A1, MYD11A1) and the surface reflectance product (MOD09A1) of MODIS were applied to estimate the daily ATI. MODIS is a key instrument aboard the Aqua and Terra whose equatorial crossing times are 1:30 am/pm and 10:30 am/pm, respectively. The spatial resolution of the MOD11A1 and MYD11A1 products is 1 km. They provide day and night LST observations at four overpass times each day. MOD09A1 includes seven bands (620–670, 841–876, 459–479, 545–565, 1230–1250, 1628–1652, and 2105–2155 nm) and has a spatial resolution of 500 m and a temporal resolution of 8 days. It was resampled to 1 km to ensure a consistent spatial resolution. MODIS data are provided by NASA Earth Science Data for free (https://earthdata.nasa.gov/).

### 2.4. Soil Texture Data

Soil texture information is used to predict sub grid SM standard deviations for the coarse-resolution satellite SM data using the MvG model, implemented in the Rosetta software [63]. The 1-km resolution soil texture data in the study area were derived from the Harmonized World Soil Database (HWSD) [64], which includes the physical and chemical properties of topsoil (0–30 cm) and subsoil (30–100 cm). This dataset has been widely used in studies of the Heihe River basin [65–67], which encompasses the study basin. The sand, silt and clay contents, and bulk density of topsoil in the HWSD were used by the MvG model. The main types of topsoil in the BRB are silty loam and loam (Figure 1b), which are mainly distributed in the valley and mountains, respectively. There is some sandy loam over the high mountains forming the basin boundaries. In general, the soil texture at low altitudes is finer than that at high altitudes in the BRB.

### 2.5. In-Situ Data

The in-situ SM data were provided by the observation dataset, known as WATER-NET, collected by the ecohydrological wireless sensor network deployed in the upper Heihe River basin through the Heihe Watershed Allied Telemetry Experimental Research (HiWATER) program launched in 2015 [49]. Eco-hydrological variables such as evaporation, runoff, soil temperature and SM were measured at a frequency of 5 min. There were

25 WATERNET sites in 2015 (Figure 1a). A few observations were missing due to a variety of factors such as weather and instrumental errors. We extracted those available SM measurements at a depth of 4 cm at each site and aggregated them into daily SM means, which was then used to check the linearity between SM and ATI in this cold alpine basin and to validate the downscaled results. We found that the in-situ SM observations at site 53 are unrealistic in value (about 0.68 cm³/cm³) and also much higher than those at other sites, likely caused by instrumental errors. Thus, site 53 was excluded in this study. The soil temperatures measured at a depth of 4 cm were also processed into daily means and used to distinguish thawing and freezing states at the sites. As the SMAP SM data are labelled as uncertain quality if the grid cell contains a small portion of freezing areas, they were especially assessed with the in-situ SM observations at the times when soil temperatures below 0 °C. Precipitation data providing background information were aggregated from the four available automatic weather stations [68] (Figure 1a) using a simple Thiessen polygon method.

## 3. Methodology

### 3.1. Formulation

If two variables $u, v$ have the same probability distribution, the standardized z-scores of these two variables will be equal (Equation (1)):

$$\frac{u_i - \bar{u}}{\sigma_u} = \frac{v_i - \bar{v}}{\sigma_v} \tag{1}$$

where $\bar{u}, \bar{v}$ and $\sigma_u, \sigma_v$ are the expectations and standard deviations of $u$ and $v$, respectively, and $u_i, v_i$ are the observations. Therefore, if exists a certain proxy variable $p$ that has the same probability distribution as SM ($\theta$) within a coarse grid cell, the value of $\theta$ can be predicted by the proxy variable. Equation (1) can be rewritten as Equation (2) :

$$\hat{\theta}_{i,j} = \bar{\theta} + \sigma_\theta(\bar{\theta}) \frac{p_{i,j} - \bar{p}}{\sigma_p} \tag{2}$$

where $\bar{\theta}$ is the SM in a coarse-resolution grid cell and $\sigma_\theta(\bar{\theta})$ is the sub grid SM standard deviation within that cell. $\bar{p}$ and $\sigma_p$ are the mean and sub grid standard deviation of the proxy variable in the same coarse-resolution grid cell, respectively. $\hat{\theta}_{i,j}$ is the predicted fine-scale SM (e.g., 1 km) at the location (i, j) within the coarse grid cell, and $p_{i,j}$ is the proxy value at the corresponding location. In Montzka's study [48], field capacity is substituted for the proxy variable of Equation (2). However, Equation (1) actually implies a linearity between the variables $u$ and $v$, as represented by Equation (3) that is equivalent to Equation (1):

$$u = a \cdot v + \beta \tag{3}$$

where $\alpha = \sigma_u / \sigma_v$, and $\beta = \bar{u} - \bar{v} / \sigma_v$. Therefore, the linearity between SM and the proxy variable should be at least maintained in every coarse grid cell, although the slope ($\alpha$) and intercept ($\beta$) in this equation can vary across grid cells. Temporally varying SM and temporally constant FC are unable to fulfil this requirement.

Previous studies [25,29,30] have discovered good linear correlation between ATI and SM at site scale. Equation (4) is thus derived by substituting the proxy variable with ATI:

$$\hat{\theta}_{i,j} = \bar{\theta} + \sigma_\theta(\bar{\theta}) \frac{a_{i,j} - \bar{a}}{\sigma_a} \tag{4}$$

where $a_{i,j}$ is the ATI value at the sub grid location (i, j), and $\bar{a}$ and $\sigma_a$ are the mean ATI and the sub grid standard deviation within the coarse grid cell, calculated by the fine-resolution ATI values within the coarse cell, respectively. Equation (4) is different from the common statistical form between SM and ATI, which assumes an invariant statistical relationship between the dependent and independent variables over the entire study area.

The linear coefficients in Equation (4) can vary across space to represent the strong spatial heterogeneity in cold alpine areas, as long as the linearity between SM and ATI is preserved in every coarse grid cell. We investigated this assumption using SM observations at the WATERNET sites before we applied this approach to downscale SMAP SM data.

### 3.2. Calculation of Subgrid SM Standard Deviations

SM generally exhibits strong spatial and temporal heterogeneity [69,70]. There is considerable sub grid variability within the coarse satellite SM grid covering hundreds of square kilometers [71,72]. Sub grid variability of SM can be obtained using SM retention curves. From the perspective of hydraulics, unsaturated soil water movement is determined by the relationship between unsaturated hydraulic conductivity ($K$) and water pressure head ($h$) in which $h$ is a function of SM ($\theta$) [73]. $K$ is hard to be directly measured because of its strong heterogeneity in space and is often estimated by SM retention curve as implemented in the Brooks-Corey model [74], MvG model [47], and Gardner-Russo model [75]. Among them, the MvG model is considered to be well capable of simulating $K$s of various soil types [46].

In the MvG model, $K$ [cm/d] is derived from the integral in Equation (5):

$$K = \Theta^{\frac{1}{2}}\left[\frac{\int_0^{\Theta} \frac{1}{h(x)}dx}{\int_0^1 \frac{1}{h(x)}dx}\right]^2 \tag{5}$$

where $h$ is the pressure head, and $\Theta$ is the effective saturation defined as $(\theta - \theta_r)/(\theta_s - \theta_r)$, where the subscripts $s$ and $r$ indicate saturated and residual values of SM ($\theta$), respectively, and $\Theta$ is given by Equation (6):

$$\Theta = \frac{1}{(1 + (\alpha \mid h \mid)^n)^m}, h < 0 \tag{6}$$

where $\alpha$ and $n$ are the parameters to be determined, and $m = 1 - 1/n$.

Combining the above equations, $K$ can be analytically solved [47]:

$$K(\Theta) = K_s \Theta^{\frac{1}{2}}[1 - (1 - \Theta^{1/m})^m]^2 \tag{7}$$

where $K_s$ is the saturated hydraulic conductivity.

Qu et al. [46] solved the sub grid SM standard deviation based on the MvG model when the mean SM of a coarse grid cell is given (Equation (8)):

$$
\begin{aligned}
\sigma_\theta^2(h) = b_0^2 \Bigg\{ & b_1^2\sigma_\alpha^2 + b_2^2\left[\frac{\sigma_f^2\rho_f}{(1 + a_2\rho_f)a_2} + \frac{a_1\sigma_\alpha^2\rho_\alpha}{(1 + a_2\rho_\alpha)a_2} + \frac{a_3\sigma_n^2\rho_n}{(1 + a_2\rho_n)a_2}\right] \\
& + b_3^2\sigma_n^2 + b_4^2\sigma_{\theta s}^2 + 2b_1b_2(-\frac{a_1\sigma_\alpha^2\rho_\alpha}{1 + a_2\rho_\alpha}) \\
& + 2b_2b_3(-\frac{a_3\sigma_n^2\rho_n}{1 + a_2\rho_n})\Bigg\}
\end{aligned}
\tag{8}
$$

where $\sigma_\theta(h)$ is the standard deviation of SM within a coarse grid cell as a function of pressure head ($h$), and $h$ can be reversed through Equation (6). $f$ is the log-transformed saturated hydraulic conductivity ($lnK_s$). $\alpha$ and $n$ are yet undetermined soil parameters. $\rho$ is the vertical correlation length of the respective parameters, and $\sigma$ represents the standard deviation. The coefficients $a_1 \sim a_3$ and $b_0 \sim b_4$ are defined as follows:

$$a_1 = \frac{(\frac{5}{2} - \frac{1}{2\bar{n}}) \cdot (\bar{\alpha}\bar{h})^{\bar{n}}}{1 + (\bar{\alpha}\bar{h})^{\bar{n}}} \cdot \frac{\bar{n}}{\bar{\alpha}} \tag{9}$$

$$a_2 = \frac{(\frac{5}{2} - \frac{1}{2\bar{n}})(\bar{\alpha}\bar{h})^{\bar{n}}}{1 + (\bar{\alpha}\bar{h})^{\bar{n}}} \cdot \frac{\bar{n}}{\bar{h}} \tag{10}$$

$$a_3 = \frac{(\frac{5}{2} - \frac{1}{2\bar{n}})(\bar{\alpha}\bar{h})^{\bar{n}}}{1 + (\bar{\alpha}\bar{h})^{\bar{n}}} \cdot ln(\bar{\alpha}\bar{h}) + \frac{ln(1 + (\bar{\alpha}\bar{h})^{\bar{n}})}{2\bar{n}^2} - \frac{2}{\bar{n}^2 - \bar{n}} \tag{11}$$

$$b_0 = (\bar{\theta}_s - \bar{\theta}_r)(\frac{\bar{\alpha}\bar{h}}{[1 + (\bar{\alpha}\bar{h})^{\bar{n}}](\bar{\alpha}\bar{h})^{\bar{n}\bar{n}}}) \tag{12}$$

$$b_1 = \frac{\bar{n}(\bar{\alpha}\bar{h})^{\bar{n}} + 1 - \bar{n}}{\bar{h}} - \frac{[\bar{n}(\bar{\alpha}\bar{h})^{\bar{n}} + 1](\bar{\alpha}\bar{h})^{\bar{n}}}{1 + (\bar{\alpha}\bar{h})^{\bar{n}}} \cdot \frac{\bar{n}}{\bar{\alpha}} \tag{13}$$

$$b_2 = \frac{\bar{n}(\bar{\alpha}\bar{h})^{\bar{n}} + 1 - \bar{n}}{\bar{h}} - \frac{[\bar{n}(\bar{\alpha}\bar{h})^{\bar{n}} + 1](\bar{\alpha}\bar{h})^{\bar{n}}}{1 + (\bar{\alpha}\bar{h})^{\bar{n}}} \cdot \frac{\bar{n}}{\bar{h}} \tag{14}$$

$$b_3 = -\frac{1}{\bar{n}} - \ln(\bar{\alpha}\bar{h}) - \ln(\bar{\alpha}\bar{h}) \frac{[\bar{n}(\bar{\alpha}\bar{h})^{\bar{n}} + 1](\bar{\alpha}\bar{h})^{\bar{n}}}{1 + (\bar{\alpha}\bar{h})^{\bar{n}}} \tag{15}$$

$$b_4 = \bar{n}(\bar{\alpha}\bar{h})^{\bar{n}} + 1 \tag{16}$$

in which bars over the parameters indicate their averages.

The topsoil properties in the 1-km HWSD dataset including sand, silt and clay contents, and bulk density were input to the Rosetta software [63], and the pedotransfer functions implemented in Rosetta estimated the parameters $(K_s, \theta_s, \theta_r, \alpha, n)$, which are required by the MvG model, for each fine (1-km) grid cell. Then, the means and standard deviations of each parameter were calculated for all SMAP grid cells. The SM means $(\bar{\theta})$ for the coarse grid cells are given by the SMAP values. The mean pressure heads $(\bar{h})$ can be computed according to Equation (6) where $\Theta$ is a function of $\theta$. The coefficients $a_1 \sim a_3$ and $b_0 \sim b_4$ were obtained following Equations (9)–(16) once the means of pressure head and the parameters $(K_s, \theta_s, \theta_r, \alpha, n)$ are known. At last, sub grid SM standard deviations for all SMAP grid cells can be solved using Equation (8).

### 3.3. Calculation of Apparent Thermal Inertia

ATI represents the relative value of thermal inertia, defined as follows:

$$ATI = C \cdot \frac{1 - a_0}{A} \tag{17}$$

where $C$ is the solar correction factor, $a_0$ is the land surface broadband albedo, and $A$ is the diurnal oscillation of temperature. A higher ATI indicates a greater resistance of soil to temperature change. In this study, ATI are estimated using MODIS LST and reflectance products.

$C$ is given by Equation (18):

$$C = \sin\varphi \cdot \sin\delta \cdot (1 - \tan^2\varphi \cdot \tan^2\delta)^{\frac{1}{2}} + \cos\varphi \cdot \cos\delta \cdot \arccos(-\tan\varphi \cdot \tan\delta) \tag{18}$$

where $\varphi$ is the latitude [rad], and $\delta$ is the solar declination [rad], defined as follows:

$$\delta = 0.00691 - 0.3999912\cos(\Gamma) + 0.070257\sin(\Gamma) - 0.006758\cos(2\Gamma)$$
$$+0.000907\sin(2\Gamma) - 0.002697\cos(3\Gamma) + 0.00148\sin(3\Gamma) \tag{19}$$

where $\Gamma$ is the day angle [rad], which can be calculated from the day of year $(n_d)$: $\Gamma = (2\pi(n_d - 1))/365.25$.

Broadband albedo ($a_0$) can be estimated by the conversion formula proposed by Liang [76]:

$$a_0 = 0.160\alpha_1 + 0.291\alpha_2 + 0.243\alpha_3 + 0.116\alpha_4 + 0.112\alpha_5 + 0.081\alpha_7 - 0.0015 \tag{20}$$

where $\alpha_1 \sim \alpha_7$ are the reflectance values of MODIS bands 1, 2, 3, 4, 5, and 7, respectively.

The diurnal temperature variation in a certain day can be estimated by the model of diurnal temperature circle curve. Peters et al. [30] approximated the diurnal temperature circle using a sinusoid defined by the following:

$$T(t_i) = \bar{T} + \frac{A}{2}\cos(\omega t_i - \Psi) \tag{21}$$

where $t_i$ is the time of day [s], $T(t_i)$ is the LST [K] at time $t_i$, $\bar{T}$ is the average LST, $A/2$ is the amplitude of the diurnal circle [K], $\omega$ is the angular velocity of Earth rotation [rad/s], and $\Psi$ is the phase angle [rad].

For a given phase ($\Psi$), there are two unknowns in Equation (21): $A$ and $\bar{T}$, which can be determined by least-square fitting when two or more observations $(t_i, T(t_i))$ are provided. The MODIS sensors onboard Aqua and Terra can provide up to four LST observations each day, which can be used for a best estimate.

The phase ($\Psi$) is computed following the method by Sobrino and Kharraz [77]:

$$\Psi = \arctan(\xi) + \pi \tag{22}$$

where $\xi$ is defined as:

$$\xi = \frac{(T_1 - T_3)\big(cos(\omega t_2) - cos(\omega t_4)\big) - (T_2 - T_4)\big(cos(\omega t_1) - cos(\omega t_3)\big)}{(T_2 - T_{34})\big(sin(\omega t_1) - sin(\omega t_3)\big) - (T_1 - T_3)\big(sin(\omega t_2) - sin(\omega t_4)\big)} \tag{23}$$

in which $T_1 \sim T_4$ are the LST observations at times $t_1 \sim t_4$.

### 3.4. Evaluation and Validation

We firstly assessed the SMAP Enhanced L3 data in the BRB in comparison to in situ SM observations from the WATERNET sites. The assessments were made from three aspects: (1) Unfrozen state: we chose the period from June to October, 2015 when the surface is totally thawed in the study area to evaluate the performance of the SMAP SM data in the unfrozen period; (2) Frozen state: SMAP attempts to retrieve SM on partially frozen cells although it stops retrieving on mostly frozen cells. We used valid data pairs when the site soil temperature at 4 cm depth ≤ 0 °C to assess the data quality of SMAP SM data influenced by partial presence of frozen ground in the SMAP grid cell. It represents the partially frozen conditions occurring before the surface becomes completely frozen; (3) Entire period: all valid data pairs consisting of site observations and corresponding SMAP SM data were used. The performance metrics include correlation coefficient (R), root mean square error (RMSE), mean absolute error (MAE) and unbiased-root mean square error (ubRMSE) [78]. Only the sites with more than 10 valid data pairs were included for assessment.

The 1-km spatial resolution daily ATI data over the study area were derived from the MODIS products. Then, the ATI time series at the observation sites were extracted to check the linear relationship between ATI and daily mean SM. The dates with missing ATI values or in situ observations were excluded from this process. Considering that the SMAP started operation in April 2015 and there are many zero readings at the WATERNET sites from December to March due to extremely low liquid water content in frozen soils, the period from April to November 2015 was chosen to validate the linearity between ATI and SM.

Restricted by the limited number of observation sites in the study area, it was impossible for us to calculate true standard deviations of SM ($\sigma_\theta$) in any SMAP grid cell using in situ SM observations. We sought an alternative way to indirectly validate Equation (8).

We first examined the spatial distributions of the estimated $\sigma_\theta$ on several representative dates. Then, we investigated three selected grid cells representative of upstream, midstream and downstream of the basin for the changing pattern of $\sigma_\theta$ as a function of mean SM ($\bar{\theta}$) and for the temporal variations. By this means, the validity of the calculated results was checked by matching them with the regular pattern of $\sigma_\theta$ that changes with $\bar{\theta}$ as reported in the previous studies [46,48].

We assessed the downscaling results from two aspects. First, the assessment was made by comparing in situ SM observations and downscaled 1-km resolution SM at the sites in terms of R, RMSE, MAE, and ubRMSE. The mean values of these metrics were also calculated while the sites at which number of data pairs is extremely low (<10) were excluded from the metric averaging [79]. It is noteworthy to mention that the scale mismatch exists between sites and 1-km grid. This mismatch together with the biases between the in-situ SM and SMAP data limits its value to directly interpret these metrics. However, if the correlation between the downscaling results and in situ observations is not worse than that between the original SMAP data and in situ observations and there are no obviously increasing errors in the downscaling results, the downscaling results would be acceptable. Two additional performance metrics (G$_{PREC}$ and G$_{RMSE}$) [80], as defined in Equations (24) and (25), were employed to further assess the gain provided by downscaling relative to the original SM data. G$_{PREC}$ and G$_{RMSE}$ measure the improvements in R and RMSE obtained from the downscaling results versus in situ observations relative to the pairs of original data and in situ observations. The values of the two G-metrics range from −1 to 1, where positive values indicate downscaling results being better consistent than the original data with in-situ observations.

$$G_{PREC} = \frac{|1 - R_{LR}| - |1 - R_{HR}|}{|1 - R_{LR}| + |1 - R_{HR}|} \tag{24}$$

$$G_{RMSE} = \frac{RMSE_{LR} - RMSE_{HR}}{RMSE_{LR} + RMSE_{HR}} \tag{25}$$

where $R_{LR}$ and $R_{HR}$ are the Rs of original data and downscaled SM against the in-situ observations, respectively, and $RMSE_{LR}$ and $RMSE_{HR}$ are the RMSE values.

Second, the errors of the downscaling results include the errors arising from the downscaling approach and the biases between the SMAP data and in situ observations. The performance of the approach can be fairly evaluated after eliminating the biases in the SMAP data from the total error. Quantile mapping bias correction methods are commonly used to correct systematic distributional biases [81–84] by calibrating the cumulative distribution function (CDF) of the modelled data into the CDF of the referenced data using a transfer function. In this study, the site-wise quantile mapping functions between in situ observations and original SMAP data were established for the sites with large systematic biases and then applied to remove the data biases from the downscaling results at those sites. As a result, the corrected downscaling results only contain errors relating to the downscaling approach and were checked with the in-situ observations for performance evaluation.

## 4. Results

### 4.1. Assessment of the SMAP Data

Figure 2 presents 118 daily means of in situ SM observations spanning April to November 2015 averaged from the 24 WATERNET sites (excluding erroneous site 53) and those of SMAP SM data from the corresponding SMAP grid cells that contains the WATERNET sites. The metrics for the unfrozen state, frozen state, and entire period are presented in Table 1. A general underestimation can be observed in the SMAP data compared with the in-situ observations, especially in late April and May. Beginning from June, the temporal variabilities in these two data were in good consistency. The SMAP data show an obvious seasonality in the mean SM in the BRB that was increasing and then

decreasing and remaining high from June to September. The in-situ observations demonstrate the same pattern, but with a maximal SM period of June through July. Both datasets suggest that SM began to sharply drop after mid-September. In the case of presence of frozen surface at sites, the agreement measured between SMAP data with the in-situ observations is similar to that in the totally unfrozen scenario using valid data pairs from June to October, 2015, as indicated by only marginal differences in terms of R, RMSE, MAE, and ubRMSE (Table 1). It generally represents a partially frozen scenario occurring before the surface on the SMAP grid cell becomes totally frozen. It thus provides some evidence that SMAP is able to retrieve liquid SM contents in the onset period of freezing with an accuracy comparable to that in the completely unfrozen state. Note, SMAP does not provide SM observations for frozen ground [60]. The SMAP grid cell containing a small portion of frozen surface is still recognized by the baseline algorithm as non-frozen ground so that the SM can be retrieved.

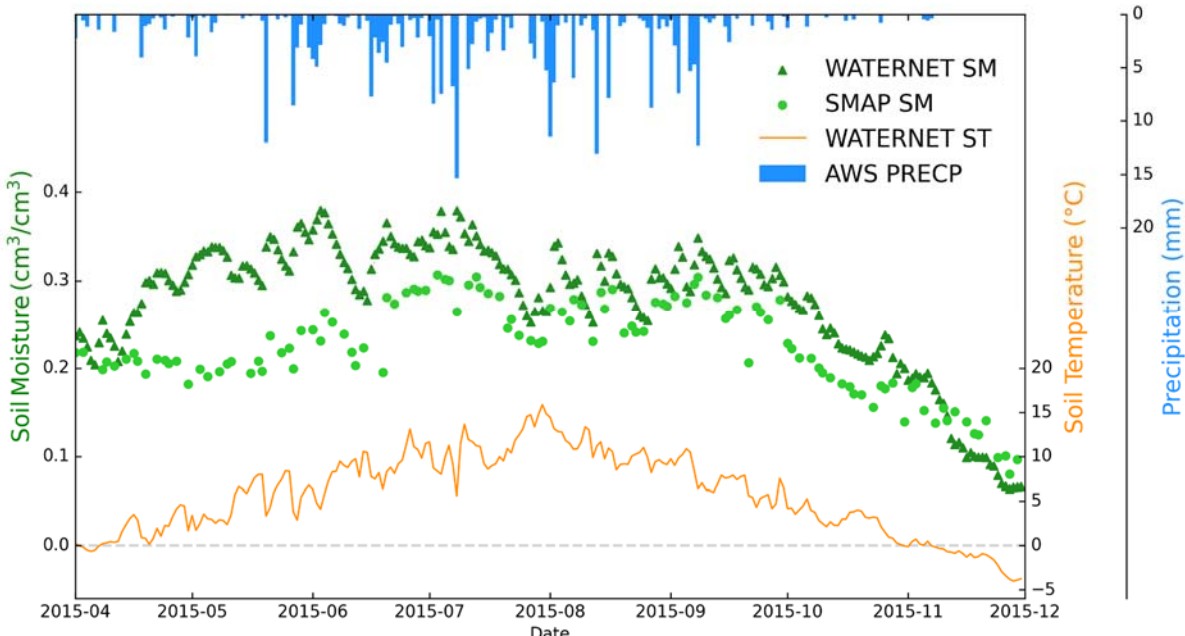

**Figure 2.** Time series of daily means of SM in 2015 from the WATERNET sites (WATERNET SM) and SMAP data (SMAP SM) in the BRB as well as precipitation (AWS PRECP) and soil temperatures (WATERNET ST) as background information. The daily means were aggregated from the observations at the WATERNET sites and the corresponding SMAP grid cells that cover the WATERNET sites. Precipitation was averaged by a Thiessen polygon method from the four automatic weather stations.

**Table 1.** Accuracy metrics measured between in situ SM observations and the corresponding SMAP Enhanced L3 data at all WATERNET site during the period of April to November 2015. R: correlation coefficient; RMSE: root mean square error; MAE: mean absolute error; ubRMSE: unbiased root mean square error; *n*: number of data pairs.

| State * | R | RMSE($cm^3/cm^3$) | MAE($cm^3/cm^3$) | ubRMSE($cm^3/cm^3$) | *n* |
|---|---|---|---|---|---|
| Unfrozen | 0.524 | 0.107 | 0.087 | 0.047 | 1345 |
| Frozen | 0.554 | 0.098 | 0.082 | 0.058 | 345 |
| Entire period | 0.527 | 0.112 | 0.097 | 0.065 | 2169 |

\* Unfrozen period: valid data pairs from June to October, 2015; Frozen period: valid data pairs when the site soil temperature at 4 cm depth ≤ 0 °C, representing the partially frozen scenario occurring before the surface becomes totally frozen; Entire period: valid data pairs from April to November, 2015.

The largest discrepancies between SMAP data and the in situ observations were found in the period of late April and early June. In this period the frozen soils and snow-pack accumulated in winter and early spring began to thaw downward as air temperature continuously increased. The great discrepancies in this thawing period contribute considerably to the low consistency in the entire period as shown in Table 1. In cold alpine areas, the onset of thawing frozen soils and snow cover may vary at locations where topography, solar radiations and land surface conditions are distinct [85]. While site observations can accurately capture those spatial-temporal SM dynamics caused by frozen soils and snow-pack thawing, the SMAP Enhanced L3 data, representing an average condition of SM in a 9 km grid, are incompetent in fully reflecting such occurrences in a cold alpine basin. Therefore, the discrepancies between SMAP data and in situ observations in the study area contain at least instrumental errors and scale mismatch.

### 4.2. Linearity between ATI and Soil Moisture

We selected the representative dates with fewer missing data during April to November, 2015, one from each month, and examined the spatial distributions of calculated ATI on these dates (Figure 3). Due to spatial data gaps in MODIS products, the corresponding ATI estimates are missing and left as blank in Figure 3. Usually, the diurnal temperature variations are relatively larger in the valleys of the BRB, indicating a weaker resistance to temperature change in the valley than other areas, and as a result, an increasing gradient of ATI from the valley towards the basin boundary was observed. This spatial pattern was more prominent in May, June, August, September, and October. The monthly distribution maps demonstrate that the ATI in the BRB increased from spring to summer and then decreased from autumn on. The basin-wide seasonal variations of ATI broadly matched in pace with the seasonal pattern of SM as shown in Figure 2.

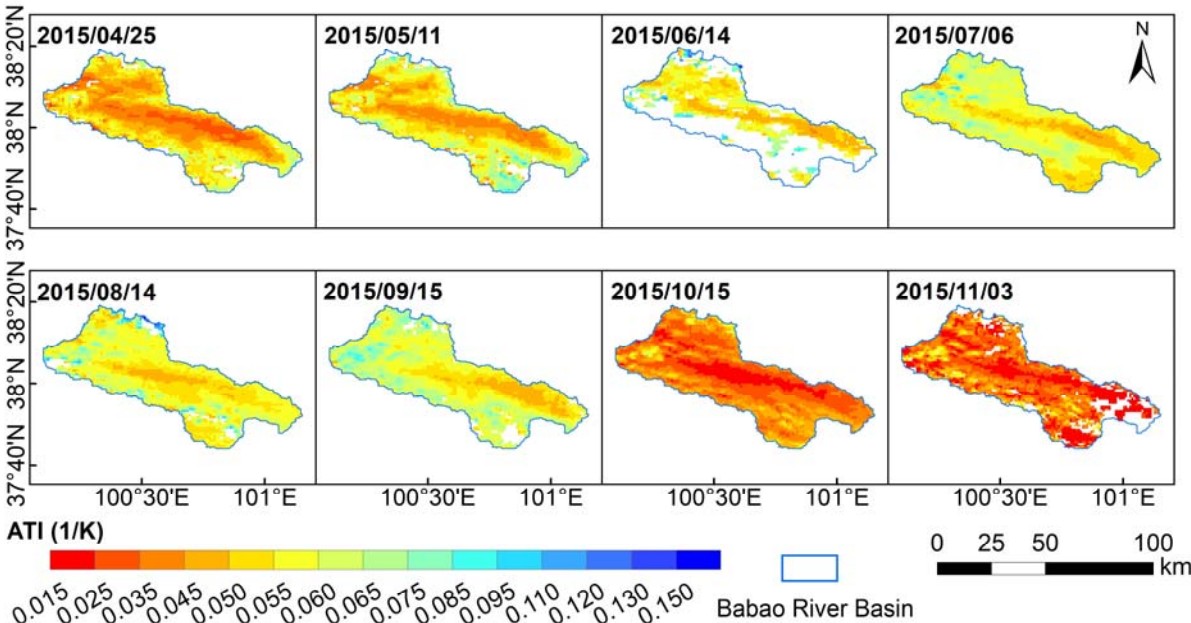

**Figure 3.** Distribution maps of the ATI on the representative dates during April to November, 2015 in the BRB.

To quantify the linearity between ATI and SM, we calculated the coefficient of determination ($R^2$) between in situ SM and ATI values at each WATERNET site (Figure 1). Significant positive correlations between the in-situ SM observations and ATI values were found at all sites (Figure 4, Table 2). Strong linearity has been observed at most sites with a mean of 0.61 in $R^2$ with all *p*-values < 0.01. This finding is in accordance with several previous observation-based studies [25,26], in which good linearity between ATI and SM

was observed with an $R^2$ of approximately 0.5–0.6, especially in low-vegetation-cover areas with a normalized difference vegetation index (NDVI) less than 0.35. The linearity worsens as vegetation cover increases [25,27]. It holds true in this study. Stronger linearity was found at sites 06, 37, and 42 ($R^2 > 0.7$, upstream) than sites 22 and 30 ($R^2 < 0.6$, downstream) as downstream areas often have better vegetation cover to relatively better thermal and hydrological conditions. In cold alpine areas, such as the BRB, vegetation is generally sparse, and even in downstream areas with relatively more vegetation cover, $R^2$ values were mostly greater than 0.5. These results confirm that good linearity between ATI and SM was maintained throughout the study area, and the linearity assumption as indicated in Equation (3) can be considered valid in cold alpine areas.

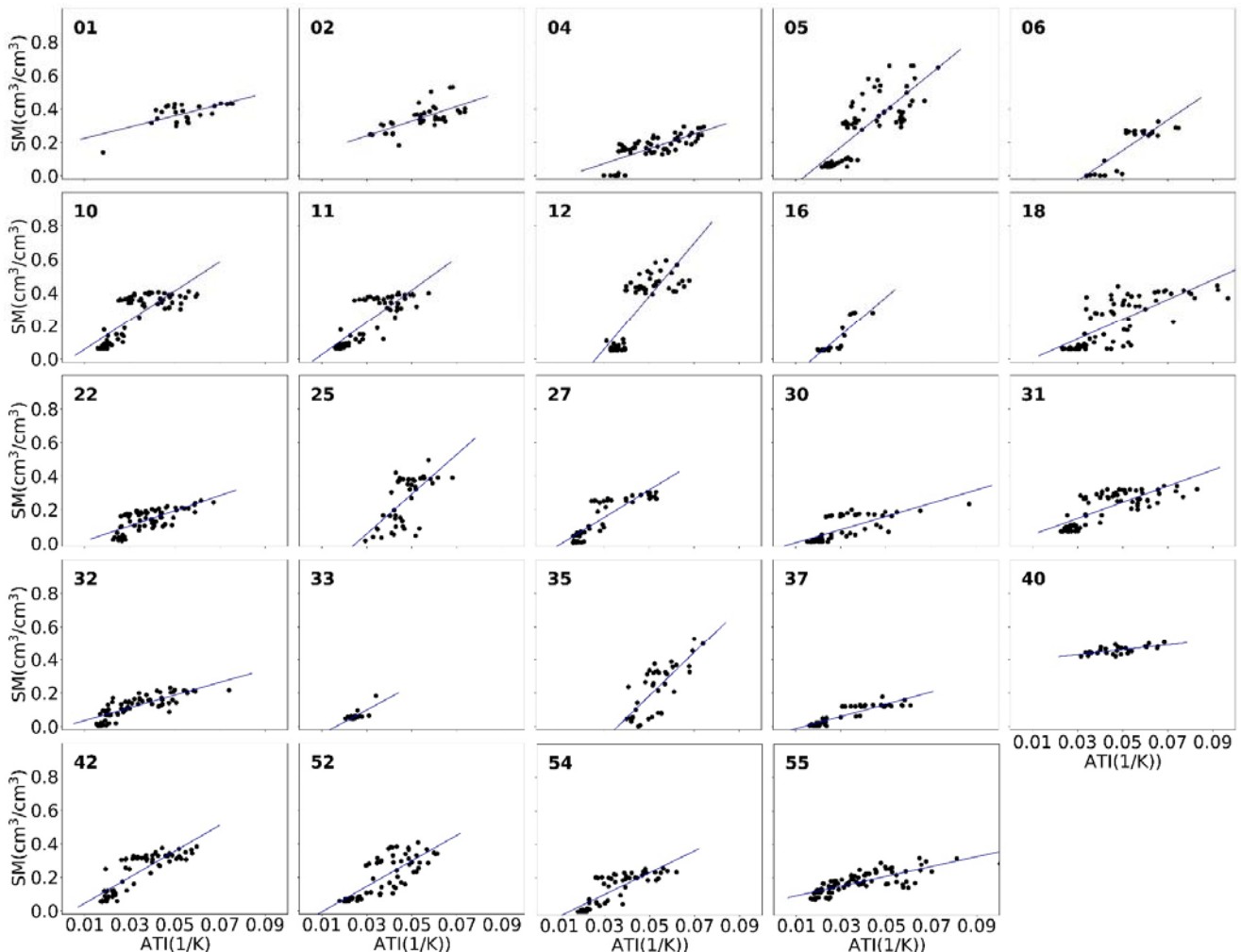

**Figure 4.** Agreement between in situ SM observations and the calculated ATI values at WATERNET sites.

**Table 2.** Coefficients of determination (**$R^2$**) between in situ SM observations and ATI values at WATERNET sites for the period of April to November, 2015.

| ID | $R^2$ * | ID | $R^2$ * | ID | $R^2$ * | ID | $R^2$ * |
|----|---------|----|---------|----|---------|----|---------|
| 01 | 0.44 | 11 | 0.69 | 25 | 0.43 | 37 | 0.81 |
| 02 | 0.49 | 12 | 0.67 | 30 | 0.56 | 40 | 0.43 |
| 04 | 0.54 | 16 | 0.76 | 31 | 0.60 | 42 | 0.72 |
| 05 | 0.65 | 18 | 0.57 | 32 | 0.64 | 52 | 0.62 |
| 06 | 0.71 | 22 | 0.51 | 33 | 0.56 | 54 | 0.67 |
| 10 | 0.63 | 27 | 0.75 | 35 | 0.57 | 55 | 0.62 |

* *p*-value < 0.01. Mean: 0.61.

### 4.3. Estimated Subgrid Standard Deviations for the SMAP Grid

According to Qu's method [46], the sub grid standard deviation of SM is expressed as a function of the mean SM of the grid cell and also affected by soil properties. A large sub grid SM standard deviation corresponds to a SMAP grid cell with strong spatial heterogeneity in soil property. Those grid cells with homogeneous soil texture have usually small sub grid standard deviations in SM (Figure 5). Figure 6 shows the SMAP SM distribution in the BRB on the same selected dates. By comparing Figure 5 and Figure 6, the SMAP grid locations with larger mean SM correspond to more SM standard deviations. This phenomenon has also been reported in several other studies [86–88]. Therefore, when the basin-wide SMAP SM first increased and then decreased through one year (Figure 6), the sub grid SM standard deviation distributions presented a similar seasonal trend (Figure 5a). In early November, when SM was low everywhere in the basin, some high standard deviations in the previous months declined in response to the decreasing SM (Figure 5a). The absence of SM standard deviations at the grid cells near the eastern boundaries on the November 03 map is caused by the data gaps in the original SMAP Enhanced L3 product.

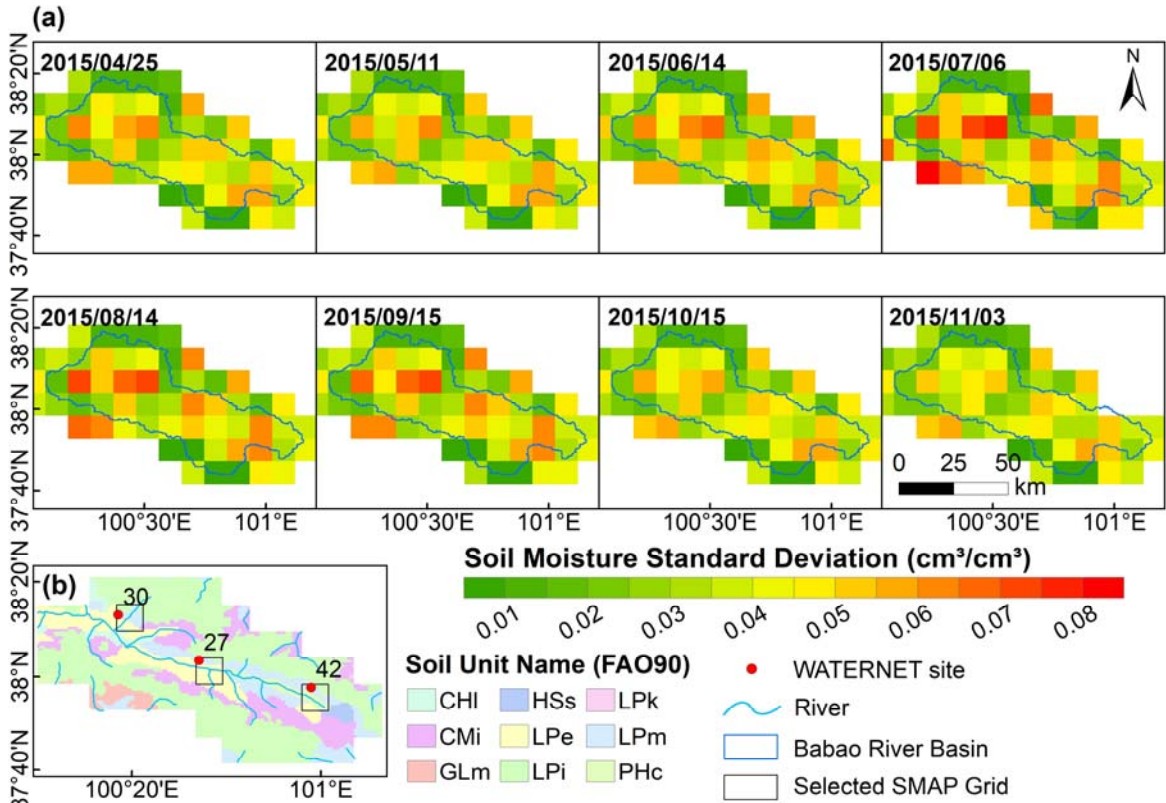

**Figure 5.** (**a**) Maps of the subgrid standard deviation of SM for SMAP grid cells on the representative dates of 2015 in the BRB; (**b**) the locations of the three selected SMAP grid cells and the HWSD soil types over the basin. CHl: Luvic Chernozems; CMi: Gelic Cambisols; GLm: Mollic Gleysols; HSs: Terric Histosols; LPe: Eutric Leptosols; LPi: Gelic Leptosols; LPk: Rendzic Leptosols; LPm: Mollic Leptosols; PHc: Calcaric Phaeozems.

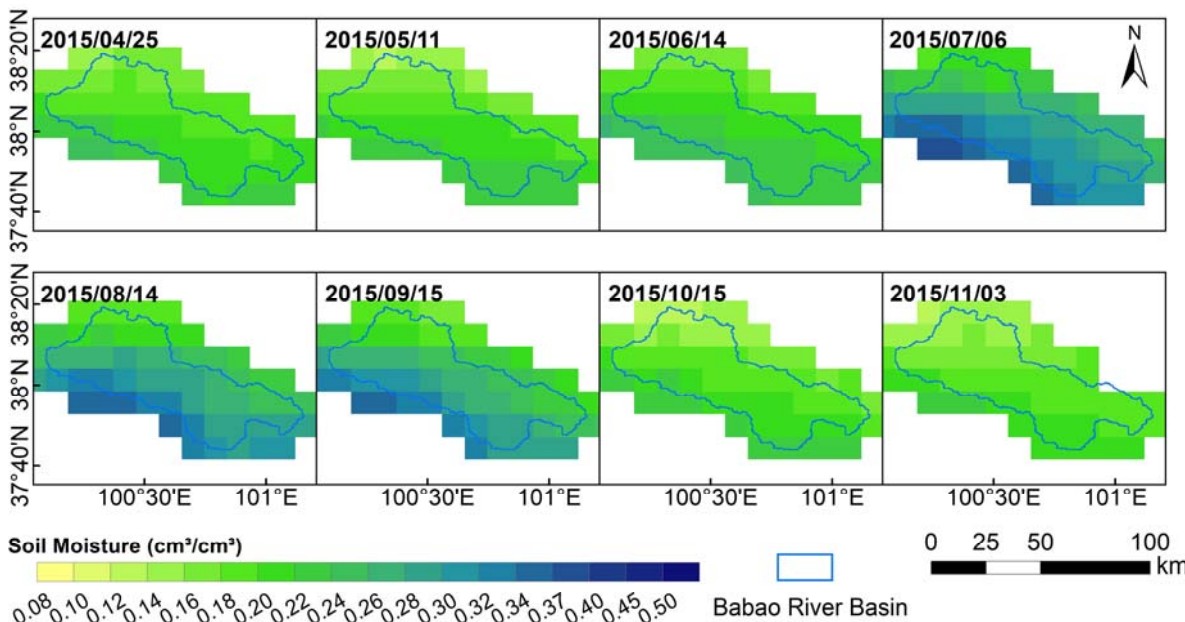

**Figure 6.** Distribution maps of the SM from the SMAP Enhanced L3 products (9-km resolution) on the representative dates of 2015 in the BRB.

The three selected SMAP grid cells after the IDs of the WATERNET sites that located within the cells have different soil properties (Figure 5b). Among them, the soil texture in grid 27, located in midstream, is more homogeneous than that in the other grid cells. There are three soil units in grid 27 and LPe (Eutric Leptosols) dominates, while the other two grid cells contain at least four soil units. As shown in Figure 7, the sub grid SM standard deviation ($\sigma_\theta$) grows as the mean SM ($\bar{\theta}$) becomes larger. Compared with the grid cells where soil properties are homogeneous, in those with more heterogeneous soil properties (e.g., grid 30 and grid 42), $\sigma_\theta$ increases more rapidly with increasing $\bar{\theta}$, and the same $\bar{\theta}$ corresponds to a higher $\sigma_\theta$, which is consistent with some other studies [48,87]. Figure 7 also illustrate the phenomenon when $\bar{\theta}$ falls within the range from 0.25 to 0.30 cm³/cm³, the changes in $\sigma_\theta$ are rather minor. Some previous studies [46,87,88] reported similar observations.

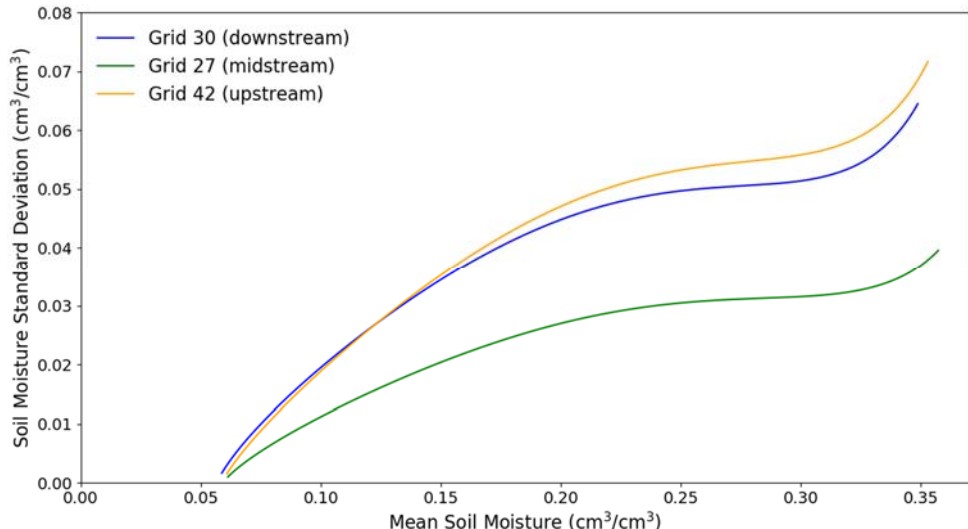

**Figure 7.** Function curves of $\sigma_\theta(\bar{\theta})$ at the selected SMAP grid cells.

The time series of $\sigma_\theta$ in the three grid cells are displayed in Figure 8a. The $\sigma_\theta$ in grid 42 was the largest, followed by that in grid 30 and grid 27. The function curves of $\sigma_\theta(\bar{\theta})$ in grid 42 and grid 30 are very similar (Figure 7). The differences in $\sigma_\theta$ between the two grid cells (Figure 8a) were mainly due to the lower $\bar{\theta}$ in grid 42 than grid 30 (Figure 8b). Although the $\bar{\theta}$ in grid 27 was the largest among these three grid cells (Figure 8b), the $\sigma_\theta$ calculated by $\sigma_\theta(\bar{\theta})$ was the smallest due to more homogenous soil properties in grid 27.

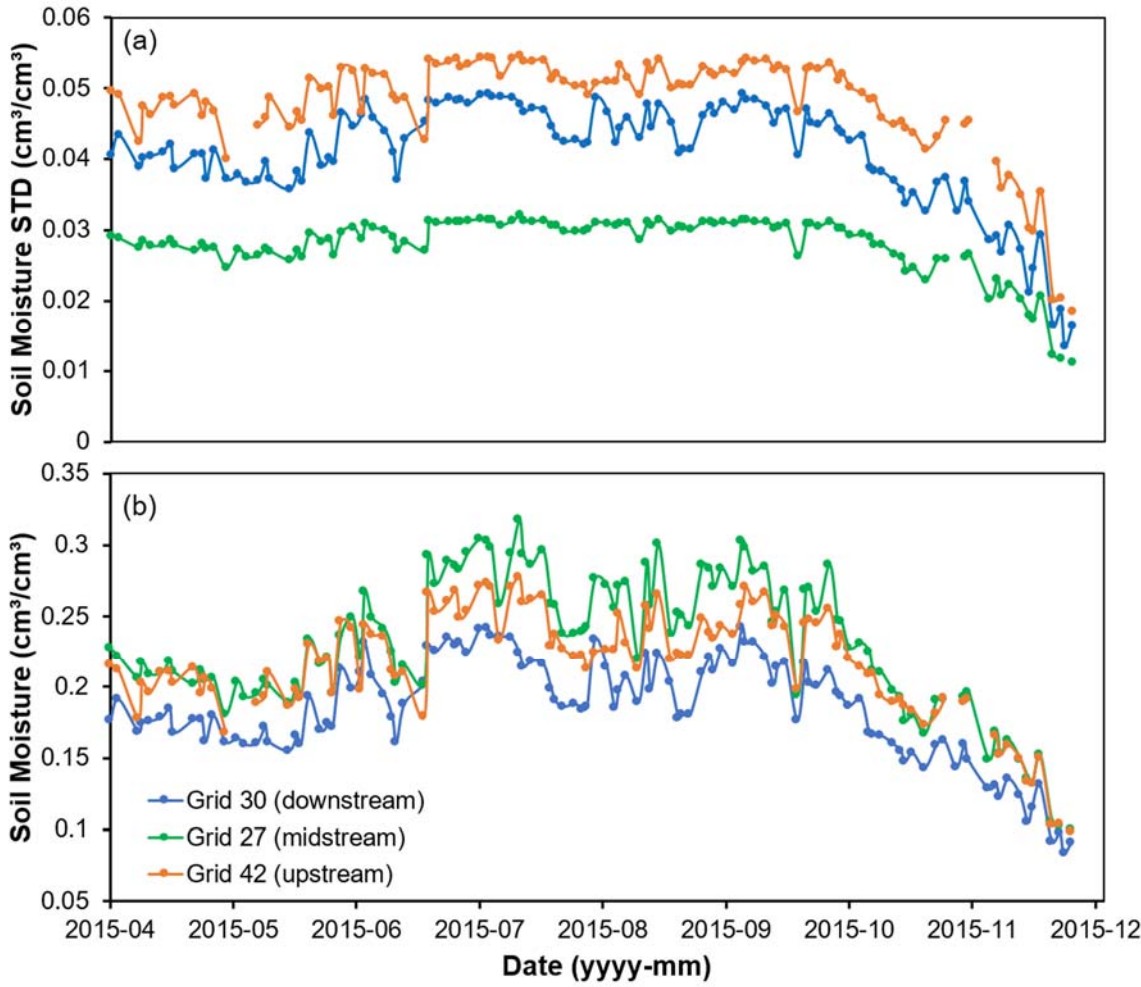

**Figure 8.** Temporal changes of (**a**) sub grid standard deviation and (**b**) the SMAP SM on the selected SMAP grid cells.

### 4.4. Validation of Downscaled Results

Figure 9 shows the downscaled 1-km grid SM maps on the representative dates of 2015. Compared with the original SMAP data (Figure 6), the downscaled maps have enriched information within the SMAP grid cells. The sharp grid borders were much less pronounced in the downscaled maps. However, the performance is still restricted by the original quality of SMAP. For instance, there were remaining horizontal edges in the northern part of the basin on August 14 and September 15. The downscaled 1-km SM maps present similar spatial patterns that SM increases with the altitude and spatially from the valleys in the center to the southern boundary (Figure 9). Those patterns are generally similar to those of ATI (Figure 3) because the spatial patterns of SM are mainly governed by orographic precipitation, topography, and soil texture within the basin. In addition, the SM conditions near the southwest boundary were much higher than those in the northeast in summer months in correspondence with the original SMAP data in those areas (Figure 6). Due to the concentrated precipitation on higher hillslopes in sum-

mer [46], wetter moisture regimes were found at high altitudes that form the basin boundary in July through September. The same vertical SM zonality has been reported by Gao et al. [89] whose work is based on in situ observations and found that the SM conditions were relatively lower in the valley of the BRB than on hillslopes. They explained with a distributed hydro-ecological model that although precipitation increases along with elevation, precipitation is mostly consumed by evapotranspiration until elevations exceeding 3200 m where evapotranspiration declines with the altitude. As a result, high SM content is sustained at high altitudes in the BRB. Those characteristics are well reflected in the spatial patterns of the downscaled SM but not in the original SMAP data.

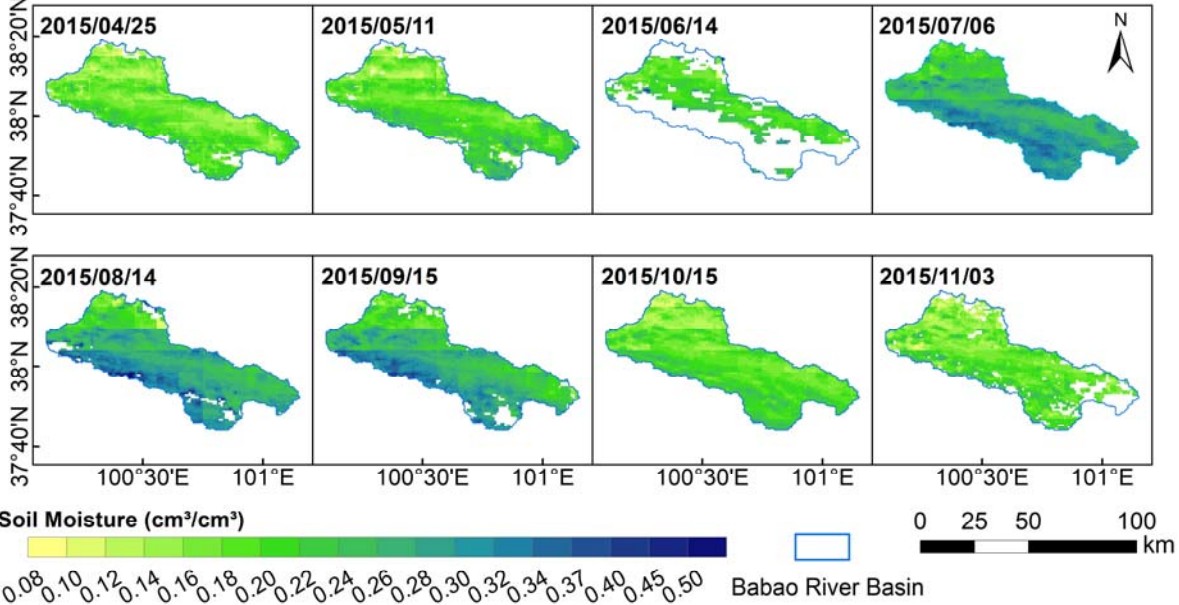

**Figure 9.** Distribution maps of the downscaled 1-km grid SM on the representative dates of 2015 in the BRB.

According to the precipitation records from the automatic weather stations in the BRB [49], continuous rainfalls occurred in the days before July 06, August 14 and September 15 (Figure 2), leading to wetter soil conditions on these dates in both the original SMAP product and the downscaling results. However, almost no rainfall occurred after September 15, resulting in an obvious decrease in SM from September 15 to October 15. In the downscaling maps, the decreases in SM were more obvious in the high-altitude hillslopes than in the low-altitude valleys, which were most likely caused by the interflow from high altitudes downstream.

In situ observations were used to evaluate the downscaling approach at each WATERNET site. Considering the missing data in the in-situ observations and downscaling results, only the periods from April to November 2015 when both of them were available were taken into comparison (Figure 10). The in-situ observations and downscaling results had good consistency and similar trends at most sites, with a mean R exceeding 0.7 (Table 3). The values of $G_{PREC}$ and $G_{RMSE}$ were positive at most sites, indicating good improvements in the downscaling results with respect to the original SMAP data. These metrics provided proof that the proposed downscaling approach produces good quality results in downscaling satellite SM data.

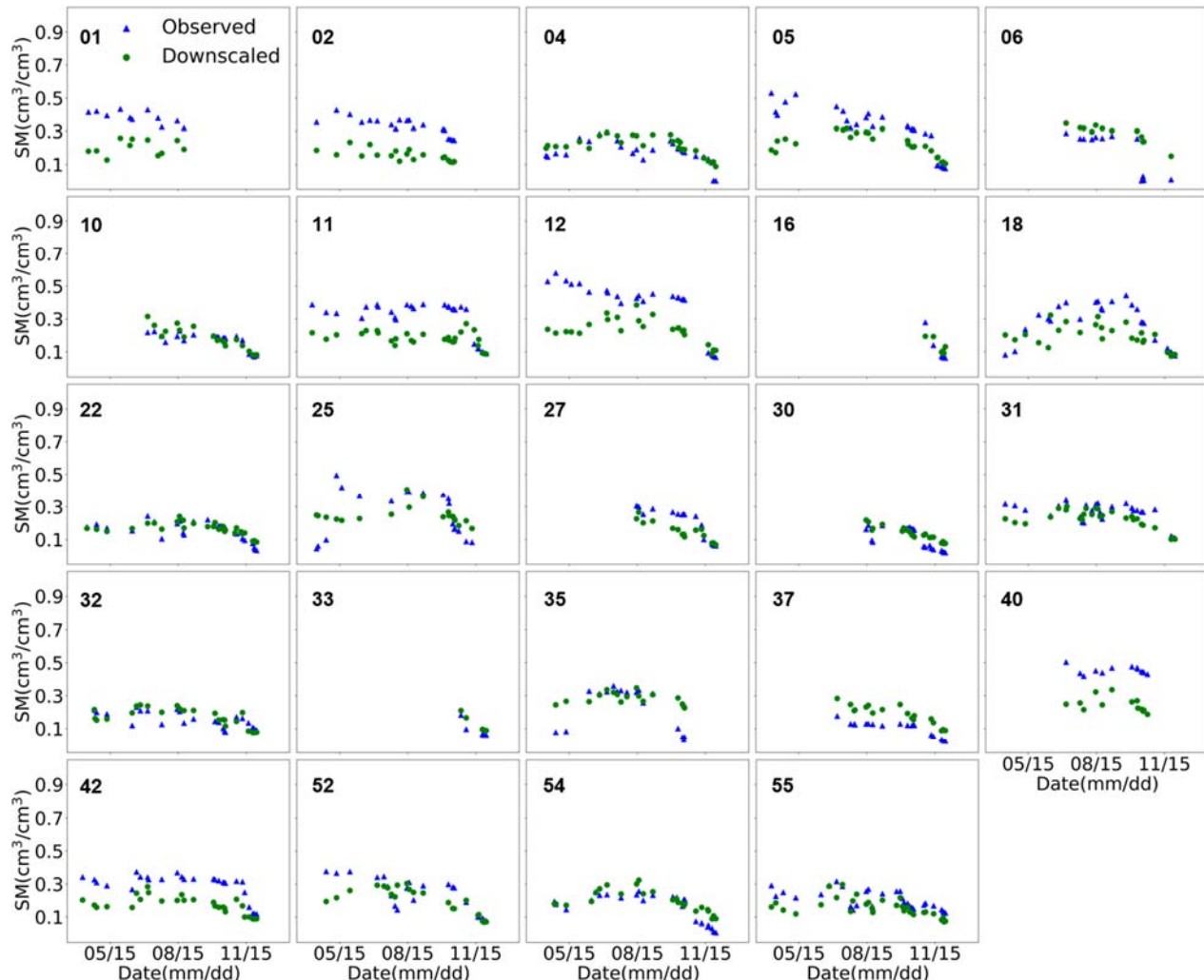

**Figure 10.** Comparisons of downscaled SMAP data and in situ SM observations at the WATERNET sites. The numbers marked are the site IDs.

However, the approach did not perform well at all sites. At some sites, the discrepancies were still pronounced, and the approach tends to underestimate SM content compared to the in situ observations. The underestimations were partly due to scale mismatch, but more importantly, they were inherited from the SMAP data that appeared to underestimate SM values at these sites (Figure 2). These biases in SMAP data may partially come from the underestimation in the surface temperature [58,59]. The downscaling results performed poorly in spring at sites 5, 12, 25, and 52 (Figure 10). It can be explained by thawing processes occurred at those site scales in spring that are hard to be represented by a coarse SMAP resolution. As shown in Figure 11, when the 4-cm soil temperatures gradually rose from below zero to positive degrees from April to early May at these sites, frozen ground and snow cover started to melt, bringing high surface SM measured at those sites. Meanwhile, those increases in SM cannot be well captured by the original SMAP data. The consequence was thus reflected in underestimating SM in the downscaling results. After the thawing ended, a good fitness came back as observed at sites 5 and 25 (Figure 10). At sites 1 and 40, underestimations prevailed throughout a year. The two sites are situated at the foothills downstream surrounded by high mountains and cannot be properly represented by the corresponding SMAP grid cells where the terrains are averaged to be high elevated. The lack of spatial representativeness of the sites led to large biases in the downscaling results (R < 0.5, RMSE > 0.18 cm³/cm³).

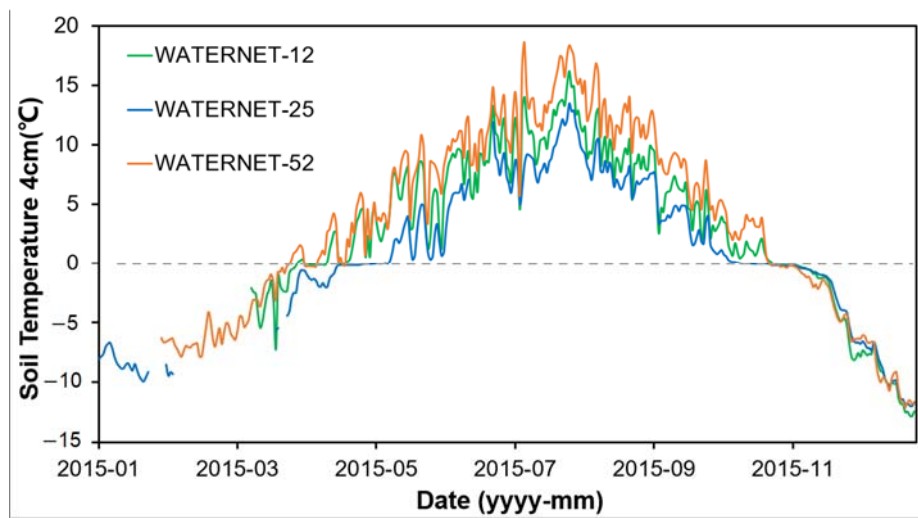

**Figure 11.** Time series of the daily soil temperature at 4 cm depth at three WATERNET sites.

**Table 3.** Accuracy metrics measured between in situ SM observations and the downscaled SM at the sites for the period of April to November 2015. R: correlation coefficient; RMSE: root mean square error; MAE: mean absolute error; ubRMSE: unbiased root mean square error; Gprec and Grmse: gain in R and in RMSE by applying downscaling [80]; *n*: number of data pairs.

| ID | R * | RMSE (cm³/cm³) | MAE (cm³/cm³) | ubRMSE (cm³/cm³) | G_PREC | G_RMSE | *n* | ID | R * | RMSE (cm³/cm³) | MAE (cm³/cm³) | ubRMSE (cm³/cm³) | G_PREC | G_RMSE | *n* |
|---|---|---|---|---|---|---|---|---|---|---|---|---|---|---|---|
| 01 | 0.235 *** | 0.189 | 0.183 | 0.050 | 0.004 | −0.076 | 11 | 27 | 0.817 | 0.078 | 0.065 | 0.052 | −0.117 | −0.159 | 15 |
| 02 | 0.669 | 0.180 | 0.175 | 0.038 | 0.470 | 0.516 | 19 | 30 | 0.746 | 0.051 | 0.046 | 0.042 | −0.187 | 0.054 | 21 |
| 04 | 0.799 | 0.051 | 0.040 | 0.040 | 0.584 | 0.513 | 28 | 31 | 0.781 | 0.054 | 0.044 | 0.040 | 0.254 | 0.173 | 27 |
| 05 | 0.712 | 0.135 | 0.110 | 0.096 | 0.361 | 0.207 | 26 | 32 | 0.680 | 0.044 | 0.038 | 0.041 | 0.196 | 0.129 | 24 |
| 06 | 0.848 | 0.133 | 0.107 | 0.079 | 0.121 | −0.336 | 12 | 33 | 0.944 | 0.040 | 0.037 | 0.016 | 0.643 | 0.208 | 5 |
| 10 | 0.818 | 0.039 | 0.029 | 0.038 | 0.147 | 0.166 | 20 | 35 | 0.743 | 0.115 | 0.085 | 0.102 | 0.250 | 0.052 | 17 |
| 11 | 0.699 | 0.152 | 0.137 | 0.086 | −0.231 | 0.233 | 28 | 37 | 0.911 | 0.086 | 0.082 | 0.025 | 0.354 | 0.080 | 18 |
| 12 | 0.610 | 0.196 | 0.172 | 0.111 | 0.622 | 0.501 | 24 | 40 | 0.433 | 0.210 | 0.206 | 0.038 | −0.078 | −0.050 | 13 |
| 16 | 0.820 | 0.054 | 0.050 | 0.050 | 0.075 | 0.310 | 8 | 42 | 0.821 | 0.128 | 0.120 | 0.043 | 0.225 | 0.323 | 27 |
| 18 | 0.681 | 0.120 | 0.103 | 0.088 | −0.228 | −0.019 | 25 | 52 | 0.658 | 0.095 | 0.070 | 0.076 | 0.164 | −0.015 | 22 |
| 22 | 0.829 | 0.032 | 0.027 | 0.031 | 0.247 | 0.042 | 29 | 54 | 0.863 | 0.052 | 0.042 | 0.042 | 0.129 | −0.090 | 25 |
| 25 | 0.438 ** | 0.128 | 0.108 | 0.127 | −0.054 | −0.009 | 19 | 55 | 0.718 | 0.060 | 0.053 | 0.041 | −0.055 | −0.083 | 31 |
|  |  |  |  |  |  |  |  | Mean † | 0.742 | 0.096 | 0.082 | 0.062 | 0.148 | 0.114 |  |

\* significance level: 0. 05. \*\* SI: 0.1; \*\*\* not statistically significant, † excluding sites 1, 40 (mis-representativeness) and sites 16, 33 (*n* < 10)

If we exclude the sites subject to mis-representativeness (sites 1, 40) and sites 16, 33 without adequate data length (*n* < 10), the means of R, RMSE, MAE, and ubRMSE between the downscaling results and the in situ observations were 0.742 (range 0.438–0.911), 0.096 (range 0.032–0.196) cm³/cm³, 0.082 (range 0.027–0.175) cm³/cm³, and 0.062 (range 0.025–0.127) cm³/cm³, respectively. For comparison, the original SMAP data have an averaged R, RMSE, MAE, and ubRMSE of 0.641 (range 0.150–0.856), 0.104 (range 0.035–0.226) cm³/cm³, 0.088 (range 0.032–0.217) cm³/cm³, and 0.062 (range 0.023–0.128) cm³/cm³, respectively. The mean G_PREC and G_RMSE were 0.148 and 0.114, respectively, which means that the downscaling results were better than the original data in terms of the gains in R and RMSE. Given the original SMAP data are at a 9-km resolution and the downscaled data are at a 1-km resolution, the metric-based comparison shows the downscaling approach did does not introduce additional errors to the downscaled results in respect to the original SMAP data. Despite large discrepancies observed in some sites, the overall favoring metric values of the proposed approach at most sites are indicative of its effectiveness in downscaling SMAP data into 1 km resolution.

To fairly assess the errors associated with the approach itself, we employed the quantile mapping technique to remove the data errors of SMAP from the total error at several WATERNET sites with large systematic biases. As shown in Figure 12, the agreements at four typical sites measured between the corrected downscaling results and in situ observations has been greatly improved compared with the uncorrected downscaling results. The RMSE and MAE of post-correction decrease obviously (Table 4). The average RMSE of the four sites decreases from 0.164 to 0.068 cm³/cm³, and the average MAE decreases from 0.151 to 0.052 cm³/cm³. This result illustrates that the accuracy of downscaled SM has been significantly improved after the systematic biases in the original SMAP data were eliminated, which further proves the effectiveness of the proposed downscaling approach.

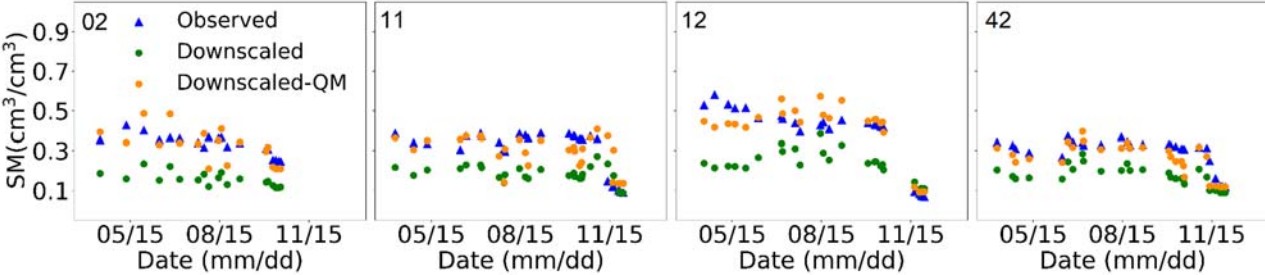

**Figure 12.** Correction of the downscaled SM data at four typical sites by removal of the original SMAP errors using quantile mapping technique. The numbers marked indicate the site IDs, Observed: in situ SM observations; Downscaled: the downscaled SM data prior to quantile mapping correction; Downscaled-QM: the corrected downscaling SM data after quantile mapping.

**Table 4.** Improved metrics (R, RMSE, MAE) of corrected downscaled SM measured against in situ SM observations at four typical sites in a period of April to November, 2015.

| ID | R * | Post−Prior | RMSE (cm³/cm³) | Post−Prior (cm³/cm³) | MAE (cm³/cm³) | Post−Prior (cm³/cm³) |
|----|-----|-----------|----------------|----------------------|---------------|----------------------|
| 02 | 0.694 | 0.025 | 0.064 | −0.116 | 0.050 | −0.125 |
| 11 | 0.612 | −0.087 | 0.087 | −0.065 | 0.065 | −0.072 |
| 12 | 0.891 | 0.281 | 0.069 | −0.127 | 0.053 | −0.119 |
| 42 | 0.863 | 0.042 | 0.054 | −0.074 | 0.041 | −0.079 |
| Mean | 0.765 | 0.065 | 0.068 | −0.096 | 0.052 | −0.099 |

* significance level: 0.05.

## 5. Discussions

Although many statistical downscaling approaches have been well established, these studies have been mostly conducted in low-altitude and flat areas where the SM distribution is relatively homogeneous and it likely exists a uniform statistical relationship between SM and proxy variables over the study area. However, cold alpine areas differ from those relatively flat areas with more complex landform, spare vegetation, cold and arid climate conditions, and presence of glaciers and frozen soils. The distribution of SM is rather heterogenous in cold alpine areas. Therefore, it's almost impossible to find a simple empirical relation between SM and environmental factors without considering the strong heterogeneity in such areas. In the BRB, precipitation shows highly non-linear relationship with elevation [50], and vegetation exhibits vertical zonality, growing better on northern slopes than on southern slopes. The spatial variability of SM is also strongly affected by snowpack and complex terrain [90]. Various meteorological and environmental factors jointly control the heterogeneous spatiotemporal distribution of SM in BRB as has been revealed by the in-situ observations and downscaling results (Figures 2 and 9). In space, SM tended to be low in the BRB valleys owing to less precipitation in valleys and

fine-grained soils impeding infiltration (Figure 1), unlike in other regions where valleys often have high SM concentration. In a cold alpine basin like BRB, SM increases with the altitude, and concentrates more on southern hillslopes than northern hillslopes. Such distribution pattern of SM in the BRB have also been reported in previous independent studies [50,51]. In time, SM increases rapidly since late April (Figure 2) due to thawing frozen soil and snowpack, until the summer when precipitation concentrates.

As an approximation of soil resistance to temperature change, ATI shows a significant relationship with the SM content in low-vegetation areas like cold alpine areas and desert [29,91]. They were found to have good linearity at individual sites in this study (Figure 4) as observed in several previous studies [25,26,30]. However, the linear coefficients varied over space and no simple statistical forms can be found to directly derive SM from ATI in cold alpine areas as does for homogenous areas. The proposed approach thus derives a statistical equation for the two variables standardized by their sub grid standard deviations, by which it takes the heterogeneity of SM into account. Semi-physical approaches are usually superior over pure statistical ones in inhomogeneous regions as demonstrated by this study. Given fine soil texture information, the hydraulic-based method of estimating sub grid SM standard deviation has proven to be sound in the BRB. It is in accordance with the theory that soil texture exerts important effects on the SM distributions on hillslopes [92], and soil hydraulic properties largely determine the shape of the SM function curve [93]. Both direct validations against in situ observations and indirect validations based on de-biased data provided proof that the proposed approach is advanced in downscaling coarse SMAP data into a finer resolution in a typical cold alpine area where common statistical methods fail.

Jin et al. [94] utilized a geographically weighted area-to-area regression kriging method to downscale the AMSR-2 SM data upstream of the Heihe River, which encompasses this study area. It produced favorite results when in situ SM observations were provided. In contrast, our approach does not depend on in situ observations and has returned satisfactory downscaling results. Considering that in situ observations are usually absent for most locations, this work therefore highlights the usefulness of the developed approach in producing fine-resolution SM data from satellite data in cold alpine areas.

Despite the success demonstrated, there are still some limitations in this study. The MODIS products often suffer missing data and invalidate ATI values at the corresponding grid cells and dates. The temporal resolution of the MODIS reflectance data is low, and the influence of terrain has not included when calculating the solar correction factor. Therefore, ATI estimates are likely subject to uncertainty. Moreover, there is a mismatch between the mean ATI and the instantaneous SMAP observation, leading to some uncertainty. It is worth noting that the relationship between SM and ATI varies in regions and seasons. Better linearity can be preserved in dry seasons and areas with low vegetation cover like this study area, but for other seasons and regions with good vegetation conditions, the relationship may degrade. In those areas, an alternative proxy variable may be expected to take the place of ATI. Consequently, this study appears to constitute a worthwhile framework for further investigation if the approach is applied to areas that have good vegetation. In addition, in the proposed approach only soil texture information is considered for accounting for SM heterogeneity. Although several studies [86,87,93] have demonstrated after testing various soil types that, soil texture data alone have provided essential information to account for SM heterogeneity using the MvG model, there is still space to be improved by including extra factors such as topography. Some studies show both soil texture and topography improve the quality of downscaling [71,72,95]. Thus, further investigation can be undertaken to include topography as an extra factor in this framework to represent SM heterogeneity in cold alpine areas.

Although we tested the approach with the SMAP data in the BRB, this approach can be applied to all satellite SM datasets. Zeng et al. [96] found the ESA CCI (European Space Agency's Climate Change Initiative) SM dataset is superior to other satellite SM data on the Tibetan Plateau. The SMAP data have been found disadvantageous of capturing SM

in spring in cold alpine areas [62]. This was reproduced in this study that during the transition period from spring to summer in late April and May, the SM variabilities in the BRB fail to be captured in the coarse SMAP grid when the SM is dramatically variable in space and time due to thawing frozen soils and snow cover. It results in a lowered final downscaling accuracy in this transition period. In fact, if a better satellite SM product is used, the proposed approach can perform even better, and this has been evidenced through the experiment in which the systematic errors in the SMAP data were removed using a quantile mapping method and the accuracy of the corrected downscaling results has improved as expected.

One distinctive characteristic in cold alpine basins is a long period when the soils are frozen. SMAP is unable to retrieve SM for frozen ground but can provide liquid SM content observations with uncertain quality for the cells where only a small portion is frozen. As we found, while the quality is acceptable in the onset period of freezing with respect to the in-situ observations, it is subject to large uncertainty in the period when intensive thawing continues to occurred in spring. Thus, the quality of downscaling for the SMAP cells influenced by partially frozen ground is mostly restricted by the SMAP SM data quality in those cells. It was reported that NASA GEOS Version 5 model can estimate both solid and liquid soil water contents in the cold alpine areas [62,97]. In this case, this approach should be modified and extended to include additional proxy variable to downscale ice contents.

## 6. Conclusions

In the present work, we extended and improved a previous work of downscaling satellite SM data based on a semi-physical approach by using ATI as a proxy variable in place of field capacity. Instead of directly deriving SM from ATI based on a conventional statistical form, this approach builds a relationship between two standardized variables of SM and ATI, where the sub grid standard deviation of SM is estimated by a hydraulic-based Mualem-van Genuchten model with known soil texture information. We re-inferred the mathematical assumption behind the approach and verified its validity in a cold alpine basin with wireless sensor network (WATERNET) installed. The approach has been applied to downscale the 9-km-resolution SMAP Enhanced L3 products into 1-km resolution in the BRB in the Qilian Mountains of Northwest China. The following conclusions can be drawn:

(1) In cold alpine areas, in situ SM observations present site-wise good linearity with the calculated ATI values, satisfying the mathematical assumption of linearity behind our approach. Similar seasonality and spatial distribution were found in SM and ATI. The mean $R^2$ between ATI and the in-situ SM observations were measured as 0.61 at all WATERNET sites in the BRB.

(2) Sub grid SM standard deviation is used to account for SM heterogeneity in the approach and they were successfully estimated by the MvG model fed with fine-resolution soil texture data.

(3) The downscaled 1-km resolution SM data showed reasonable spatial and temporal patterns in the BRB and well agreed with in situ SM observations, with an average correlation coefficient of 0.742 and small RMSE, MAE and ubRMSE values. After removing systematic errors contained in the original SMAP data from the downscaled results reassessing the performance showed better metric values, further confirming the effectiveness of the downscaling approach.

Overall, the use of the proposed semi-physical approach has proven satisfactory in downscaling the satellite SM data in cold alpine areas. The resulting fine-resolution SM data can serve as useful databases for land surface and hydrological studies in cold alpine areas.

**Author Contributions:** Conceptualization—Z.N. and H.G., methodology and formal analysis—H.G., Z.C., Y.Z. and Z.N., resources—Z.N., supervision—Z.N., writing—original draft—Z.C., H.G., and Z.Y., writing—review and editing—Z.N. and Z.C. All authors have read and agreed to the published version of the manuscript.

**Funding:** This study was supported by National Key R&D Program of China (2017YFA0603603), National Natural Science Foundation of China (41931180, 41971074 and 41671055).

**Institutional Review Board Statement:** Not applicable.

**Informed Consent Statement:** Not applicable.

**Data Availability Statement:** The ATI, downscaled 1km SMAP SM, and the uncorrected and corrected data associated with Figure 12 presented in this study are openly available at https://doi.org/10.6084/m9.figshare.13669960. The in-situ SM observations at the WATERNET sites are available at https://doi.org/10.11888/Soil.tpdc.270896.

**Acknowledgments:** The authors would like to express their gratitude to National Tibetan Plateau Data Center (http://data.tpdc.ac.cn) for providing the WATERNET soil moisture observations, National Snow & Ice Data Center (https://nsidc.org) for providing SMAP soil moisture data, and NASA (https://earthdata.nasa.gov) for providing MODIS data. The Harmonized World Soil Database used in this study was collected from National Cryosphere and Desert Data Center (http://www.crensed.ac.cn/portal/). We are very grateful to the editor and anonymous reviewers.

**Conflicts of Interest:** The authors declare no conflict of interest.

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
