# Peer review of "A Semi-Physical Approach for Downscaling Satellite Soil Moisture Data in a Typical Cold Alpine Area, Northwest China"

_remotesensing, doi:10.3390/rs13030509_

Round 1
Reviewer 1 Report
This paper presents a method for downscaling surface soil moisture, which extends the early work by Carsten. Even though this method targets a cold alpine area in Northwest China, it should be widely useful to many mountainous headwater areas with complex terrain, strong soil heterogeneity, and orographic precipitation. The presentation of this paper is in good detail but could be briefer especially the method introduction. The result discussion is concrete for demonstrating the strength and limits of this proposed method. Overall, this is a solid study. It is interesting and useful to the research community relying on reliable soil moisture data. I suggest a possible publication after fixing minor issues, as stated in my comments.
Specific comments:
- I suggest the authors give a clear definition of “surface soil moisture (SM)" at the beginning of the introduction section, especially the surface soil layer's thickness. This is very important to readers because the surface layer could have different thicknesses in different applications. With a clear specification of surface layer thickness, it makes more sense to discuss the SM influences to other variables, such as energy and water partitions.
- In line 53, the authors mentioned “strong penetrability” and “allows observations in all weather”. I am not clear about the “strong penetrability” of microwave remote sensing. Besides, does microwave remote sensing well detect soil moisture in all weather conditions? Please clarify.
- Please provide citations for the summary “Most study areas in SM downscaling research have low altitudes and are relatively flat.” (Line 107)
- Please provide brief reasons for “cold alpine areas are characterized by strong land-surface heterogeneity and high nonlinearity between SM and environmental variables” comparing with other regions. (Line 108).
- In Line128 and Line129, please find a better word to replace “unfortunately”. This word is slightly emotional and may not fit a search paper.
- In Line 143 and 144, does the “continental river” mean “inland river”?
- In section 2.2, please describe the detection depth of the SMAP data. In section 2.3, please describe the detection depth of the MODIS data. If the two depths are not the same, please describe how to connect the two datasets.
- For equation (2), my understanding is that ? bar is the soil moisture of the coarse-resolution grid, and it is treated as the mean of sub-grid soil moisture. The underlying assumption of this downscale approach should be that the soil moisture of the coarse-resolution grid is the mean soil moisture of sub-grid cells inside it. Is my understanding correct? If yes, “the mean SM of the coarse-resolution grid” should be “the SM of the coarse-resolution grid”. Similarly, in equation (5), ? bar should be the ATI of the coarse-resolution grid. For each time instance, you only have one value of ATI, it doesn’t make much sense to emphasize “mean” in my opinion.
- I suggest the authors using “grid” and “cell” separately. Maybe you can call the coarse-resolution box a grid and call the fine-resolution box a cell. If “grid” and “cell” is defined clearly, it will be easier for both the authors and the readers.
- The subtitle “3.2 Calculation of standard deviations for SMAP girds” is a little bit misleading to me. My understanding is “Calculation of standard deviations for coarse-resolution grids”, while the “coarse-resolution grids” are the same as “SMAP grids”. When you use “SMAP grids”, I would associate it with the sensor errors of SMAP. Also, in Line 253, please correct “girds” with “grids”.
- For section 3.3, I suggest the authors more explicitly emphasize that the ATI is directed from MODIS data products.
- In Line 319 and Line 320, the authors stated that there were many zero readings at WATERNET sites from December to March due to the extremely low SM. Is it right? I guess the reason is that the ground was frozen and the liquid part of SM is extremely low. However, it may not be accurate to claim “extremely low SM”, since ice is also a part of SM.
- In equation (25), what is the reason to use the absolute value of 1-R_LR, 1-R_HR, 1-R_LR, and 1-R_HR? R_* are correlation coefficients, 1-R_* is non-negative anyway. It seems that there is no need to use the absolute value operators “||” in the equation if my understanding is correct.
- Please provide more details or citations for the quantile mapping method described from Line 354 to Line 360.
- Are there any contributions of ground freezing on the sharp drop of SM after September? If yes, this paper should state more clearly that the SM is the liquid part of soil moisture.
- In Figure 2, the mean SM on 10/15/2015 is smaller than that on 09/15/2015. Why are the ATI values on 10/15/2015 generally higher than those on 09/15/2015 in Figure 3?
- In 409, if the “P” in “P<0.001” represents the “p-value”, it is better to use a lower case “p” in italic font. This is the same for Figure 4.
- This paper has employed the value of the SMAP data. Is there any consideration to use the temporal variables? In soil moisture data assimilation, the anomaly of SMAP data is used more than the magnitudes for avoiding the impacts of systematic bias. I suggest the authors exploring the possibility of downscaling satellite-derived soil moisture data based on anomaly indices in future studies.
- As many land surface models can simulate ground freezing and thawing, it will be more interesting to extend this method to a partially frozen time period for the cold alpine areas in future studies.
Reviewer 2 Report
Paper review on “A semi-physical approach for downscaling satellite soil moisture data in a typical cold alpine area, Northwest China”.
The paper extended the previous work of Carsten et al. (2018) and proposed a coarse-scale satellite soil moisture downscaling method based on a semi-physical approach forming from a statistical relationship between standardized soil moisture (SM) and apparent thermal inertia (ATI). A key assumption of this method is that there is a linear relationship between SM and ATI, and this assumption has been tested in three WATERNET sites from upstream, midstream, and downstream of the Babao River basin, a typical cold alpine area in China. The proposed method was applied to downscale the SMAP enhanced passive SM products from 9 km to 1 km, and the downscaled SM was validated by using ground measurements from 25 WATERNET sites. The results show though the downscaled SM generally underestimate ground observations, they can well capture the SM variation trend. The advantages and limitations of the proposed method were also discussed. The paper is generally well organized and easy to follow, and it deserves to be published in ‘Remote Sensing’ with the following comments to be addressed appropriately.
General comments:
- As mentioned by the authors that most previous studies downscale coarse-scale satellite SM in flat and homogeneous regions, and thus they focus on the heterogeneity issue in the proposed SM downscaling method. The heterogeneity of SM is caused by many factors, such as precipitation, topography, soil texture, and land cover. In this study, the sub-grid standard deviation of SM is estimated by the MvG model along with soil texture information. Therefore, it seems that only the soil texture heterogeneity is considered in their method. From Fig. 1(a), it can be seen the topography also exerts large heterogeneity. Is there any way to consider these heterogeneous factors (may be in the future) in the calculation of SM standard deviation?
- There is freeze-thaw process in the study area as mentioned in the manuscript. Theoretically, the microwave satellite cannot obtain reliable SM during frozen seasons. Thus, when using the SMAP SM products, the regions of snow and ice, and frozen ground should be discarded (e.g., see Entekhabi et al., 2010, doi: 10.1109/JPROC.2010.2043918). It is not clear if the authors have used data only in unfrozen seasons (e.g., surface temperature > 0 degree) or in both frozen and unfrozen seasons. This issue should be clarified in the manuscript.
- In the study, the RMSE, MAE, R, Gprec and Grmse were used to assess the accuracy of downscaled SM. I suggest the authors add the unbiased RMSE (ubRMSE) which considers the additive mean bias due to the scale mismatch between satellite data and ground measurements (see Entekhabi et al., 2010, doi: 10.1175/2010JHM1223.1). This error metric has been extensively used to evaluate the performance of satellite SM products in many previous studies.
Specific comments:
- Line 28: please add the unit for RMSE.
- Line 29 to 31: the characteristics of precipitation, evapotranspiration and runoff in the BRB are not presented in the paper.
- Line 58: delete “accuracy”.
- Line 172-178: the authors actually used the SMAP enhanced passive SM products, while the validation references here are all for SMAP passive SM. The evaluation of SMAP enhanced passive SM can be found in many previous studies (e.g., Chan et al., 2018, http://dx.doi.org/10.1016/j.rse.2017.08.025; Chen et al., 2018, doi: 10.1109/TGRS.2017.2762462; Cui et al., 2018, doi:10.3390/rs10010033).
- Line 179: is it because the surface temperature is below zero degree?
- Section 2.4: if there is more than one site in a satellite grid, then the averaged measurements were used?
- Section 3.2: it is better to summarize the input and output of the MvG model for the calculation of SM standard deviations, which will be clearer for the readers.
- Line 262: change “Mualem-van Genuchten (MvG) model” to “MvG model” since the full name of MvG has already been mentioned in line 120.
- Line 285-286: check the character (i.e., theta) here since it seems not consistent with that in equation 7.
- Line 323-325: how many sites (the most sites) in one SMAP grid?
- Line 367: how many SMAP grids?
- Figure 3: are the spatial data gaps in Figure 3 due to the effect of cloud?
- Figure 4: does the linear relationship between ATI and SM hold for other sites?
- Line 428-429: any possible reasons for this phenomenon?
- Line 445-456: how did you confirm that soil texture in grid 27 is more homogeneous than that in the other grids?
- Line 451-453: it seems strange, are there any possible reasons for this phenomenon?
- Figure 9: it is better to use the same SM range for figure 9 and figure 5.
- Line 515-519: the underestimation of SMAP SM has already been found in previous studies and some of them found the underestimation of the surface temperature partially contributes to the dry bias of SMAP SM (see Chen et al., 2018; Cui et al., 2018). Did the authors validate the surface temperature in SMAP product by using ground measurements in the Babao River basin?
- Line 542-543: what are the error metrics for the original data?
- Line 558-568: please give more details (also cite the corresponding references) regarding the bias-correction method used here. Brocca et al. (2013, Scaling and Filtering Approaches for the Use of Satellite Soil Moisture Observations) have summarized some bias-correction methods for satellite SM products, and it seems that the authors adopted other method. Moreover, it is not clear why the R is improved after bias-correction since the commonly used bias-correction methods do not affect the temporal trend of the original data.
- Line 633-646: the authors can try to downscale the ESA CCI SM in the future since the ESA CCI SM is a daily SM and previous study (Zeng et al., 2015, http://dx.doi.org/10.1016/j.rse.2015.03.008) has found the ESA CCI SM is superior to other satellite SM in the Tibetan Plateau which is also a cold alpine area similar to the Babao River basin.
Reviewer 3 Report
The paper presents a semi-physical method for downscaling 9km SMAP soil moisture to 1km over a cold alpine area in China. The method is based on the relationship between apparent thermal inertia, obtained from MODIS, as proxy of actual thermal inertia, and soil moisture. as the soil moisture sub-pixel variability is required, the within-pixel standard deviation is obtained from soil texture information.
I have a few major concerns:
1. it seems that no pre-processing and data cleaning was performed on neither the satellite SM nor in-situ observations (e.g. removing observations taken with temperature < 3°C), thus including observations notoriously suspicious and unreliable.
2. the proposed methodology can be potentially of interest for regions with similar environmental conditions, however, the conclusions drawn should be supported by more data and results. For instance, the choice of considering only 3 in-situ stations (instead of all the available ones) for showing the linearity between ATI and SM is ambiguous.
Specific comments:
l 19: why is the soil moisture "standardized" here?
l 57: why only AMSR-E and SMOS are mentioned, when the study focuses on SMAP? Also, one could cite that high-resolution SM products can be obtained through, e.g., Sentinel-1 but in this case the temporal resolution is a big issue
l 146: I think that commas should be replaced by dots (2,452 --> 2.452)
l 147: I think that BRB appears for the first time here
Section 2.2: consider including quality flags for temperature, snow cover, ..
Section 2.4: SMAP soil moisture product is representative of the top ~5 cm, while the soil texture data refers to the 0-30cm layer. Could this be a problem, i.e. how much texture variability is expected in the soil profile?
Section 3.1: equations 3 and 4 are merely repetitions of Eq 2, consider removing
l 256: references needed (e.g. Zappa et al 2019 RS, Alemohammad et al 2018 HESS)
l 262: introduce the MvG acronym already in line 196
l 367: how many data pairs?
l 374: dropped --> drop
Figure 2: add precipitation and temperature to the plot. The datetime format in the x-axis is incorrect. Typo in the caption ".. soil SM .."
l 400: such a sentence should be supported by a more quantitative analysis!
Figure 3, 9: why choosing dates with MODIS missing data? It is impossible to get an idea of the behaviour in e.g. July and November
l 406: why performing the analysis to 3 sites only instead of all available ones? the conclusions would be much more robust!
Figure 7: the results shown here are quite contradictory with literature. In particular, it is assumed a convex relationship between mean SM and standard deviation (e.g. Famiglietti et al 2008 WRR, Brocca et al 2010 WRR)
l 464: this is exactly what one would expect because of how the standard deviation was calculated, i.e. related to soil texture characteristics (which are static) and soil moisture (which is dynamic)
l 497: remove "had"
l 510: provide (at least the range of) the number of data pairs
Figure 10: extreme and unrealistic values should be removed (e.g. SM > 0.7 at site 01) before the analysis! Also, for some stations the number of data pair is extremely low (site 16, 33 ,..): a minimum threshold should be applied, otherwise the site should be disregarded from the final metric averaging (e.g. Gruber et al 2020 RSE)
Figure 12: same y-axis range as in Figure 10. It would also be interesting including in the same plots the downscaled SMAP before bias removal
l 576: some discussion was already given in the Results section (for instance l 140)
l 579: many methods also consider both soil texture and topography (e.g. Alemohammad et al 2018 HESS, Montzka et al 2018 RS, Shin and Mohanty 2013 WRR, Zappa et al 2019 RS)
l 582: "There is usually no single controlling factor for SM". This statement seems to contraditct the results obtained and presented, which do not consider any of the other factors mentioned (e.g. precipitation, topography, vegetation, ..). This requires additional discussion
l 635: usually SMAP compares better than most of the other remotely sensed-derived soil moisture products (e.g. Al-Yaari et al 2019 RSE, Chen et al 2018 RSE)
l 637: "SMAP data was inferior in capturing SM .." inferior to what??
l 646: not clear what is meant
l 661: consider rewriting as bullet points for more clarity
Round 2
Reviewer 2 Report
The authors have addressed most of my previous comments and the manuscript has been substantially improved. The paper needs only minor revision before publication.
1) In the revision, the SMAP SM data were evaluated in unfrozen and frozen periods. In my understanding, SMAP has already excluded the soil moisture retrievals in frozen soils in the released product based on SMAP team’s screening criteria (e.g., when the surface temperature is below zero, the soil moisture retrievals are discarded). Therefore, there are many missing values in SMAP soil moisture data in low temperature conditions (there are no SMAP value (NaN value) in your study area from January to March, right?). However, since the surface temperature used in SMAP may be not very accurate in some areas, there will be still some soil moisture values in frozen soils (that is to say, the surface temperature used in SMAP algorithm is larger than zero while the actual soil temperature is below zero). You can try to extract the surface temperature in the SMAP data at 6:00 AM (descending) and 6:00 PM (ascending) and the corresponding soil moisture values for inspection. You may find that when the surface temperature is below zero, there will be no soil moisture value in this case. It is OK that the authors retain the comparison of the soil liquid water in frozen and unfrozen periods, but I think the above-mentioned issue should be explicitly chaired and explained clearer (after examination). Otherwise, the reader may mistakenly think that the SMAP soil moisture product contains the value in all frozen period.
2) Add the N (number of data pairs) in Table 1.
3) In the response, the authors stated "good consistency between in situ observation and SMAP L3 products were observed under the frozen states (Ma et al., 2016; Zhang et al., 2017)". The Ma et al. (2016)’s paper did not compare the in situ observations and SMAP L3 products. Did you want to cite Ma et al. (Remote Sens. 2017, 9, 327; doi:10.3390/rs9040327)? Please check the reference in the manuscript carefully.
4) It seems that the authors adopted the Cumulative Distribution Function (CDF) matching technique (i.e., the quantile mapping technique mentioned in the paper) to correct the bias of the downscaled SM. The CDF matching technique has also been applied in many previous studies, such as the merging of ESA CCI product (see Liu et al., 2012, https://doi.org/10.1016/j.rse.2012.03.014). It should be noted that the CDF matching method aims to correct the systematic bias between the original data and reference data, but not to improve the R value of the original data. It is believed that the CDF matching method will not influence the temporal trend of the original data. Therefore, it is expected the R value of the original data may not change (much) after using CDF matching technique (though the R value may slightly change). However, from Table 4, it seems the R value of downscaled SM at ID 12 is much improved. So I suggest the authors recheck their results (particularly for ID 12) carefully, and focus more on the improvement of RMSE rather than that of R in the content.
Author Response
"Please see the attachment."
